

# Spatial heterogeneity of soil organic matter and microbial community composition across ice-wedge polygons and soil layers in Arctic lowland tundra

Victoria Martin[1,2,3], Cornelia Rottensteiner[1,2,3], Hannes Schmidt[1], Moritz Mohrlok[1,3], Julia Horak[1], Carolina Urbina-Malo[1], Julia Wagner[4,5,6], Willeke A`Campo[4], Luca Durstewitz[4], Niek Jesse Speetjens[7,8], Rachele Lodi[9], Bela Hausmann[10,11], Michael Fritz [12], Gustaf Hugelius[4,5], Andreas Richter[1,2]

[1]Centre for Microbiology and Environmental Systems Science, University of Vienna, Vienna, Austria

[2]APRI, Austrian Polar Research Institute

[3]Doctoral School in Microbiology and Environmental Science, University of Vienna, Vienna, Austria

[4]Department of Physical Geography, Stockholm University, Stockholm, Sweden

[5]Bolin Centre for Climate Research, Stockholm University, Stockholm, Sweden

[6]Department of Ecology and Environmental Science, Umeå University, Umeå, Sweden

[7]Department of Earth and Climate, Vrije Universiteit Amsterdam, Amsterdam, Netherlands

[8]School of Environmental Science, University of Victoria, Victoria, Canada

[9]Institute of Polar Science, National Research Council, Venezia Mestre, Venice, Italy

[10]Joint Microbiome Facility of the Medical University of Vienna and the University of Vienna, Vienna, Austria

[11]Department of Laboratory Medicine, Division of Clinical Microbiology, Medical University of Vienna, Vienna, Austria

[12]Department of Permafrost Research, Alfred Wegener Institute Helmholtz Centre for Polar and Marine Research, AWI, Potsdam, Germany

*Correspondence to*: Victoria Martin (victoria.sophie.martin@univie.ac.at)

**Abstract**

Permafrost soils are highly vulnerable to climate change. Yet, carbon-flux forecasts for ice-wedge polygon tundra ecosystems remain uncertain due to pronounced spatial heterogeneity at both terrain and pedon scales. In this study, we investigated how soil organic matter pools, microbial community structure, and potential enzymatic activities vary across two spatial dimensions: polygon geomorphology (low-, flat-, and high-centered polygons) and soil layers (organic topsoil, mineral subsoil, cryoturbated material, and upper permafrost).

Polygon-specific signatures of SOM and microbial profiles persisted across all layers, and layer- specific effects were consistent across polygon morphologies. Low-centered polygons differed markedly from the other polygon types, exhibiting lower bioavailability of organic matter, smaller microbial abundance, and reduced potential for hydrolytic degradation. Organic topsoils were most distinct from mineral subsoils in their SOM composition and from permafrost in their microbial community structure. They also functioned as microbial hotspots, showing the



highest abundances and enzyme activities. Once thawed, permafrost SOM may also become rapidly mobilized
due to its quantity, composition, and considerable potential for hydrolytic degradation.
Taken together, our findings suggest that gradients in organic matter and redox conditions structured the variations
found at both spatial scales. Anticipated polygon transitions, active-layer deepening, and abrupt thaw with climate
change, are therefore likely to interactively accelerate soil carbon losses. We propose that distinguishing low-
centered polygons from other polygon types, and organic topsoils from deeper soil layers, provides a tractable
framework for scaling soil processes across the spatially heterogeneous Arctic lowland tundra.

## 1 Introduction

Permafrost-affected landscapes are characterized by high surface and sub-surface variability (Ping et al., 2015;
Siewert et al., 2021). Over centennial time scales, periglacial processes have formed a dynamic mosaic of different
geomorphological landscape elements in close spatial proximity (Washburn, 1956). Ice-wedge polygons are
among the most widespread ones in continuous permafrost regions (French, 2007a; Washburn, 1973). Covering
approximately one third of the Arctic landmass, they are particularly prevalent in ground-ice rich lowland tundra
and thermokarst ecosystems of Siberia and North America (Brown, 1967; Fritz et al., 2016). Cyclic freeze and
thaw events and repeated soil frost cracking have led to the development of ice wedges in the ground (French,
2007a; Washburn, 1973). Depending on the state of these ice wedges, different polygonal patterns eventually
emerge at the terrain surface through physical self-organization processes (Krantz, 1990; MacKay, 2000). When
ice wedges grow, the plastic deformation of overlying soil strata results in elevated rims that enclose lower located
areas, forming so-called low-centered polygons (LCPs) (French, 2007a; Washburn, 1973). Conversely, when ice
wedges degrade or the accumulation rates of overlying sediment or peat layers exceed their growth rates, so-called
high-centered polygons (HCPs) emerge (French, 2007b). Characterized by a raised mound that is surrounded by
troughs, HCPs hence show inversed topographic features to LCPs (French, 2007a; Washburn, 1973). Flat-centered
polygons (FCPs) represent an intergrade type between the two mentioned ones (Shur et al., 2025)and have
intermediate attributes such as a flat center surrounded by drainage channels (Vaughn and Torn, 2018).
The morphology of ice-wedge polygons influences soil hydrological and thermal dynamics, dictates soil type and
texture, shapes the composition of microbial and vegetational communities, and affects soil processes and the
energy balance of the ecosystem (Lara et al., 2018; Liljedahl et al., 2016; Nitzbon et al., 2019; Wainwright et al.,
2015). As a result of their microtopography, HCPs, for instance, have well-drained centers with dry surface
conditions, while the centers of LCPs regularly experience inundation and ponding (Boike et al., 2008; Nitzbon
et al., 2019). Due to higher thermal conductance in wetter soils, summer active layer depths in the polygon center
reach often deeper in LCPs than HCPs (Liljedahl et al., 2016; Speetjens et al., 2022; Walvoord and Kurylyk,
2016). Water logging strongly shapes the soil structure in LCPs. Low oxygen availability restricts decomposition
and facilitates organic matter accumulation (Donner et al., 2012; Kuhry et al., 2020), leading to the buildup of a
prominent organic layer (Organic Cryosoils). Soils in LCPs are thought to experience the least pronounced
seasonal temperature fluctuations of all polygon types (Hubbard et al., 2013), due to the combined insulating
effects through summer inundation, preferential snow accumulation during winter (Abolt et al., 2018), and peat
accumulation (Grosse et al., 2011). In the less insulated FCPs and HCPs, frost penetrates deeper, which causes





the mixing of soil layers and leads to the burial of poorly decomposed organic matter from the topsoil into the
mineral subsoil via cryoturbation (Turbic Cryosoils) (Ping et al., 2008; Wild et al., 2016).
Differences in soil properties across polygon types are closely mirrored by corresponding changes in the structure
and function of microbial and plant communities (Chu et al., 2011; Taş et al., 2018; Wolter et al., 2016). Dry areas
of HCPs are typically dominated by dwarf-shrubs, forbs, and lichen (Speetjens et al., 2022; Wainwright et al.,
2015). In LCPs, the vegetation is adapted to water-saturated conditions and graminoids or peat- and brown-mosses
prevail (Minayeva et al., 2018). Shifts in redox conditions are strong drivers of soil microbial communities and
determine potential SOM decomposition pathways (Ernakovich et al., 2017). Predominantly aerobic communities
are found in FCPs and HCPs, whereas anaerobic pathways are common in LCPs (Chowdhury et al., 2021; Frank-
Fahle et al., 2014). Due to tight interconnections between vegetation, microbial communities, and biogeochemical
cycling (Islam et al., 2020; Joabsson and Christensen, 2001; Wallenstein et al., 2007), a substantial portion of the
observed spatial variability in tundra carbon exchange may be also ascribed to different polygon morphologies
(Arora et al., 2019; Wainwright et al., 2015). For example, LCPs are considered a significant source of $CH_4$ while
in HCPs the efflux of $CO_2$ prevails (Lara et al., 2015; Sachs et al., 2010).
Another important dimension of spatial variability in permafrost-affected landscapes occurs at the pedon-scale.
Several physicochemical characteristics, such as temperature, redox conditions, ice- and organic matter content,
or bulk density exhibit pronounced vertical stratification along the soil profile. Most notably, the seasonal thaw
of the active layer contrasts sharply with the persistently frozen permafrost below. Frost heave and cryoturbation
introduce additional fine-scale irregularities in the sequence and spatial arrangement of soil horizons (Siewert et
al., 2021), each differing markedly in terms of abiotic conditions and biotic characteristics. These dynamics along
a permafrost soil profile carry important implications for the life of associated microbial communities.
For example, topsoils are subject to pronounced diurnal and seasonal temperature fluctuations, whereas deeper
soil layers remain thermally relatively stable (Baker et al., 2023; Barbier et al., 2012). The permafrost table acts
as a physical barrier for water, nutrient, and gas exchange between the active layer and frozen permafrost (Wilhelm
et al., 2011). It is however temporally variable and depending on summer conditions, the uppermost part of the
frozen permafrost (transient layer) may thaw on decadal timescales (Shur et al., 2005). In frozen permafrost, liquid
water only occurs in brine channels, and oxygen availability is also limited (Altshuler et al., 2017; Gilichinsky et
al., 2003). Within the active layer, soil texture, bulk density, and organic matter content change markedly from
the organic topsoil to the mineral subsoil, affecting also the distribution of water and oxygen (Alexander, 1989;
Bauer, 1974). In tundra ecosystems, the influence of plants also decreases rapidly with depth. About 96 % of the
root biomass is located in the top 30 cm of the soil profile (Iversen et al., 2015; Jackson et al., 1996). Only few
plants, e.g., sedges, have a deeper rooting system that affects the oxygenation, exudation, or litter inputs of deeper
layers (Joabsson and Christensen, 2001; Shaver and Cutler, 1979). Decreasing plant influence with increasing
depth leads to strong shifts not only in the quantity of soil organic matter, but also in its quality and stoichiometry
(Weintraub and Schimel, 2003). At the same time, soil pH may rise due to a reduced input of acidifying
compounds, such as those from sphagnum mosses or root-derived organic acids (Clymo and Hayward, 1982;
Jones, 1998; Vives-Peris et al., 2020). Although conditions are partly harsh and highly variable in permafrost
soils, its large habitat heterogeneity provides a wide range of ecological niches for supporting specialized



microbial communities (Malard and Pearce, 2018; Taş et al., 2018) with diverse metabolic strategies (Jansson and
Taş, 2014; Tveit et al., 2013).
Ultimately, the spatial heterogeneity of tundra ecosystems governs the interplay among soil properties, microbial
communities, and vegetation, thereby shaping soil processes and modulating the impacts of climate change.
However, biogeochemical models at the ecosystem scale are often limited by high variability at finer spatial
scales(Sturtevant and Oechel, 2013). While many studies have focused on either landform units (e.g., (Lara et al.,
2015; Liljedahl et al., 2016; Sachs et al., 2010; Wainwright et al., 2015)) or the importance of soil layers for
edaphic characteristics, microbial communities, or greenhouse gas fluxes (e.g., (Kuhry et al., 2020; Lynch et al.,
2023; Müller et al., 2018; Schnecker et al., 2015; Wild et al., 2013, 2016), relatively few have considered both
dimensions concertedly (Lipson et al., 2015; Taş et al., 2018). A more integrative characterization of permafrost
soils across both the terrain and the pedon scale would therefore not only deepen our understanding of lowland
tundra ecosystems but may also improve predictions of their future trajectories under climate change.
This study aimed to portray edaphic and microbial characteristics in the spatially heterogeneous permafrost soils
of Arctic lowland ice-wedge polygon tundra. Specifically, we asked: How do physicochemical properties, soil
organic matter composition, microbial community structure, and potential extracellular enzyme activities vary
across polygon morphology (low-, flat-, and high-centered polygons) and soil layers (organic topsoil, mineral
subsoil, cryoturbated material, upper permafrost)? Which edaphic and organic matter features direct microbial
and enzymatic patterns across scales? Do polygon morphology and soil layer influence soil processes primarily
through main effects or through their interaction? By linking patterns across both spatial scales, we aimed to
identify unifying drivers that shape soil organic matter pools and microbial communities. Such mechanistic
insights may improve predictions of the permafrost carbon–climate feedback by enabling scalable representation
of tundra heterogeneity in spatially explicit ecosystem and land-surface models.
**2 Materials and Methods**
*2.1 Study area*
We studied Arctic lowland ice-wedge polygon tundra, located on the coastal plain of the Yukon, Western Canada,
(Fig. 1). The first focus area comprised two small lagoons called Ptarmigan Bay (69°27'N, 139°05'W) and Whale
Bay (69°25'N, 138°59'W). The second focus area, approximately 40 km further towards the west called Komakuk
Beach (69°35'N, 140°10'W), is a small coastal catchment positioned between two alluvial fans (Fritz et al., 2012).
The periglacial landscape in this ecosystem is characterized by a mosaic of ice-wedge polygon networks, mires,
beaded streams, and thermokarst lakes (Fritz et al., 2012; Quaternary Geology Yukon Coastal Plain, Yukon
Territory-Northwest Territory; Speetjens et al., 2022), underlain by continuous permafrost with a high ground ice
content (Couture and Pollard, 2017; Westerveld et al., 2023). The climate is classified as Polar Tundra (Beck et
al., 2018), and the vegetation as bioclimatic subzone E/ low Arctic shrub tundra (Walker et al., 2005).
Microtopography and relief are strong determinants for the identity of the prevailing soil suborder, and plant
species composition. Turbic Cryosols were present in the drier centers of HCPs and FCPs (Canadian System of
Soil Classification, Soil Classification Working Group, 1998), where also dwarf-shrubs, forbs, and lichens
dominated the flora (Supplementary Table 1(a)). FCPs were mainly characterized by graminoid tussocks and



dwarf-shrubs. Inundated centers of LCPs harbored organic Cryosols. The dominant plant groups were graminoids,
brown mosses, and peat mosses (A guide to the landscape of the Firth River Valley, Ivvavik National Park;
Quaternary Geology Yukon Coastal Plain, Yukon Territory-Northwest Territory; Walker et al., 2005). A more
detailed description of the study area, their surface geology, glaciation history, climate, soil suborders and
vegetation, can be found in the Supplementary, and in (Wagner et al., 2023).

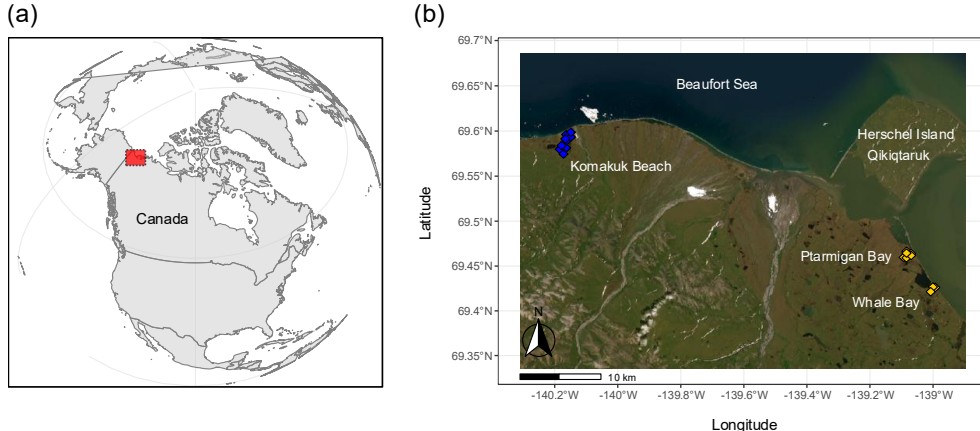


**Figure 1. Study area along the Yukon Coast, Canada (a) and areal overview of the sampling locations (b).** During the
2018 field campaign, samples were retrieved from Ptarmigan Bay and Whale Bay (yellow). Komakuk Beach (blue) was
investigated during the 2019 field campaign. Maps were created using code available on GitHub (Irwin, 2021). Basemap: (Esri,
156 2024).

*2.2 Soil sampling and sample storage*
Soil sampling was conducted during two field campaigns. Ptarmigan Bay and Whale Bay were sampled in August
2018, and Komakuk Beach in August 2019. Sampling took place in late summer, when active layer depths
typically approach their seasonal maximum. In the field, we identified larger networks of low-centered (LCP),
flat-centered (FCP), and high-centered polygons (HCP) and selected six polygons of each type for sampling
(Supplementary Table 1 (a)). For sampling the active layer in HCPs and FCPs, we excavated 1 m – 2.5 m wide
soil pits until the permafrost table was reached. We recorded active layer depths, in situ soil temperatures at a
distance of 10 cm (Supplementary Table 1(b)), classified soil horizons according to (Field Book for Describing
and Sampling SoilsNo Title), and documented their distribution and thickness (Supplementary Fig. 1). We
collected 100–200 g of fresh material from each horizon by compositing subsamples from several positions within
the soil profile. Organic horizons and cryoturbated material were sampled by cutting blocks of known dimensions
using a knife. For mineral subsoils we inserted steel cylinders (5.5 cm diameter) horizontally into the exposed
profile. Excavating soil pits was not possible for the mostly waterlogged LCPs. We therefore restricted active
layer sampling to retrieving two replicate cores per plot using a gas-powered SIPRE corer (diameter 7.5 cm).
Documentation, identification and sampling of soil horizons were done in the same manner as described for HCPs
and FCPs. In all types of polygons, we sampled the frozen part of the permafrost by using a gas powered SIPRE
corer, or by hammering a steel pipe (diameter 4.2 cm) into the ground with a sledgehammer (Hugelius et al.,
2010). For each core, we recorded the identity and dimensions of respective horizons and estimated visible ice



contents. As within this study only the upper 10 cm of the extracted permafrost cores were used, we strictly refer
to the transient layer when discussing characteristics of the permafrost layer. For more details see Supplementary.
In total, 81 soil samples were collected (Ptarmigan Bay & Whale Bay n=39; Komakuk Beach n=42). Samples
were grouped by polygon type (LCP_n=20, FCP_n=32; HCP_n=29), and by soil layer category (organic
topsoil_n=35 including O, Oi, Oe, Oa horizons; mineral subsoil_n=14 including B, Bg horizons, cryoturbated
material_n=13 including Ojj, Oijj, Oajj, Ajj horizons, and upper permafrost_n=19 including Off, Bff, Cff
horizons). Due to natural heterogeneity in the field and soil-pit specific differences in soil horizon development,
an imbalanced sampling design emerged (organic: LCP_n=12, FCP_n=12, HCP_n=11; mineral: LCP_n=2,
FCP_n=6, HCP_n=6; cryoturbated: LCP_n =0, FCP_n =7, HCP_n =6; permafrost LCP_n=6, FCP_n =7, HCP_n
=6).
Within 24 h after sampling, we carefully removed visible roots, green litter, and coarse solid organic matter
fragments from active layer samples and homogenized them by hand. Aliquots for DNA extraction were preserved
with RNAlater™ Stabilization Solution (ThermoFisher Scientific) and stored at 4 °C. Active layer samples were
stored and transported at 4 °C, and permafrost samples frozen. The samples arrived approximately two weeks
after each respective sampling campaign at the University of Vienna and were processed immediately. Prior
analysis, frozen permafrost samples were carefully thawed for two days at 4 °C. The samples of both field
campaigns were treated with the same protocols, analyzed by the same methods and combined into one dataset.
*2.3 Physicochemical soil parameters and nutrient pools*
The samples were analyzed for pH (ultra-clean water) and gravimetric water content (80 °C for 48 h). We
measured total soil Carbon (Soil C), Nitrogen (Soil N), plus their isotopic composition by an elemental analyzer
(EA 1110, CE Instruments, Italy) coupled to a continuous-flow isotope ratio mass spectrometer (IRMS, DeltaPlus,
Finnigan MAT). Following a modified ignition method (Kuo, 1996) to convert organic phosphorous (P) to
inorganic P, soil total P (Soil P) was determined photometrically in 0.5 M $H_2SO_4$ extracts via malachite-green-
assay (D'Angelo and Crutchfield, 2001). Dissolved organic carbon (DOC) and total dissolved nitrogen (TDN)
concentrations were quantified in 1 M KCl extracts via TOC/TN-Analyzer (Shimadzu, TOC-VCPH/CPNTNM-1
analyzer). For more details see Supplementary.
*2.4 Chemical composition of soil organic matter*
The chemical composition of soil organic matter (SOM) was characterized by Pyrolysis-Gas
Chromatography/Mass Spectrometry (CDS Pyroprobe 6200, CDS Analytical coupled to Pegasus BT, LECO; with
the polar column Supelcowax ™ 10 Fused Silica Capillary Column, 30 m x 0.25 mm x 0.25μm film thickness,
Sigma Aldrich), using the semi-automated approach that is described in (Martin et al., 2024) with minor
modifications. For the qualitative investigation of the SOM pool, we performed Principal Component Analysis
(PCA) on center-log-ratio (clr) – transformed abundances (mg C $g^{-1}$ DW) of 534 pyrolysis products. We further
grouped these pyrolysis products into six SOM compound groups (aromatics and phenols, carbohydrates, lignins
and lignin-derived compounds, lipids, N-containing substances, compounds of general and unknown origin;
Supplementary Table 3) and explored differences in their absolute and relative abundances among polygon types
and soil layer categories. For more details see Supplementary.



***2.5 Soil microbial communities - DNA extraction, amplicon sequencing, digital droplet (dd)PCR***
Microbial DNA was extracted (250 mg FW soil from the organic topsoil layer and 400 mg FW soil from all other
soil layers) using the FastDNA™ SPIN Kit for Soil (MP Biomedicals, Santa Ana, USA). We followed the
manufacturers' instructions but added minor modifications for the removal of the RNAlater™ Stabilization
Solution. Extraction blanks were included and subjected to subsequent quantification and sequencing steps.
Amplicon sequencing and raw data processing was performed at the Joint Microbiome Facility of the Medical
University of Vienna and the University of Vienna (JMF project ID JMF-2008-5). A two-step barcoding approach
was employed to generate amplicon libraries of archaeal, bacterial, and fungal communities using Illumina MiSeq
(V3 Kit) in the 2 x 300 bp configuration (Pjevac et al., 2021). We used the primer pairs 515F
(GTGYCAGCMGCCGCGGTAA, (Parada et al., 2016) and 806R (GGACTACNVGGGTWTCTAAT, (Apprill
et al., 2015) for amplifying the V4 hypervariable region of the 16S rRNA gene and the primer pairs ITS1F
(CTTGGTCATTTAGAGGAAGTAA, (Smith and Peay, 2014) and ITS2 (GCTGCGTTCTTCATCGATGC,
(White et al., 1990) for amplifying the fungal ITS1 region (amplification conditions in Supplementary). Amplicon
pools were extracted from the raw sequencing data using the FASTQ workflow in BaseSpace (Illumina) with
default parameters. Demultiplexing was performed with the python package demultiplex (Laros JFJ,
github.com/jfjlaros/demultiplex), allowing one mismatch for barcodes and two mismatches for linkers and primers
(Pjevac et al., 2021). Amplicon sequence variants (ASVs) were inferred using the DADA2 R package applying
the recommended workflow (Callahan et al., 2016b, a). FASTQ reads 1 and 2 were trimmed at 150 nt with allowed
expected errors of 2 (16S rRNA gene) and 230 nt with allowed expected errors of 4 and 6 (ITS1 region),
respectively. Bacterial and archaeal ASV sequences were classified using SINA version 1.6.1 (Pruesse et al.,
2012) and the SILVA database SSU Ref NR 99 release 138.1 (Quast et al., 2013) using default parameters. Fungal
ASVs were classified using DADA2 and the UNITE general FASTA release for eukaryotes (v.8.2), using default
parameters (Abarenkov et al., 2020). Datasets were deposited in the NCBI Sequence Read Archive under
BioProject accession number (PRJNA1274918).
Digital droplet PCR (ddPCR) was performed to quantify 16S rRNA genes and ITS1 regions with the same primers
used for sequencing. Each ddPCR reaction had a volume of 22 µL and consisted of 1x QX200 ddPCR EvaGreen
Supermix (BioRad), 0.1 µmol L$^{-1}$ of each primer and 0.5 ng of template for the quantification of 16S rRNA genes
or ITS1 regions, respectively. Droplets were generated on a QX200 ™Droplet Generator (BioRad) and directly
subjected to PCR amplification (amplification conditions in Supplementary Table 5). PCR products in droplets
were kept at 4 °C over night to increase their separation before measuring their fluorescence intensity (on a QX200
™ Droplet Reader, BioRad). Gene copy numbers were calculated using the QX ONE Software Standard Edition
(v. 1.2, BioRad) where thresholds between positive and negative droplet populations were set consistently for
each sample using histograms as a guide. We expressed final ddPCR results as 16S rRNA and ITS1 gene copy
numbers g$^{-1}$ DW soil and used them as abundance proxies for bacteria and archaea, and fungi, respectively.
We calculated abundances of individual ASVs (gene copy number corrected reads g$^{-1}$ soil DW) by multiplying
the 16S rRNA or ITS1 gene copy numbers measured in ddPCR assays with their respective relative abundances
from the amplicon sequencing datasets. Rare (bacterial, archaeal, and fungal) taxa (containing less than 0.05 % of
all gene copy number corrected reads per sample) were excluded from the dataset, resulting in 3643 bacterial, 137
archaeal, and 1604 fungal ASVs being considered in final analyses respectively. For investigating the microbial



community composition (β-diversity), we performed Principal Component Analysis (PCA) on gene copy number
corrected reads g$^{-1}$ DW (center-log-ratio (clr) - transformed(Aitchison, 1984). We explored quantitative
differences of certain phyla between polygon types and soil layer categories using ddPCR-corrected reads g$^{-1}$ DW.
We assessed α-diversity as richness (number of observed ASVs) and Shannon diversity. We therefore used
unfiltered, but rarefied count-datasets of bacterial and archaeal, and fungal reads. For more details see
Supplementary.

### 2.6 Microbial extracellular enzymatic activity

We measured the potential activities of six hydrolytic extracellular enzymes involved in carbon-, nitrogen-,
phosphorus-, and sulfur-cycling: β-D-1,4-cellobiosidase (exoglucanase), β-D-1,4-glucosidase (glucosidase), β-
1,4-N-acetyl-glucosaminidase (exochitinase), acid phosphatase, leucine-aminopeptidase (protease) and sulfatase,
using microplate fluorometric assays as described in (Canarini et al., 2021). For more details see Supplementary.

### 2.7 Statistical analyses and data visualization

All analyses were performed in R Studio Version 4.1.2 (R Core Team, 2017, version 4.1.2). Significances of
relationships were tested against a $p < 0.05$ threshold. Plots were generated using ggplot2 (Wickham, 2016) and
partly edited using Inkscape (Inkscape, 2020). The data is accessible under:
https://doi.org/10.5281/zenodo.17158574.
We employed linear-mixed-effects models (lmes) to test all univariate variables for the fixed effects of 'ice-wedge
polygon type' and 'soil layer category' plus their interaction. Therefore, we used the packages lme4 (Bates et al.,
2015), lmerTest (Kuznetsova et al., 2017), emmeans (Lenth et al., 2022), and car (Fox and Weisberg, 2019). Due
to the sites' very similar landscape, climate, soil, and vegetation, we determined the random effect in the lme
model as specific soil pit ID blocked within the sampling site. Model results were inspected using the anova()
function with the default being a type III analysis of variance (ANOVA). In the case of no interactive effect being
observed we used type II ANOVA to account for potential effects of different treatment replicates (Langsrud,
2003). We used the Estimated Marginal Means post hoc test to perform multiple comparisons (p.adjust='tukey')
on the fixed effects of polygon type and soil layer category. In the case of an interactive effect being observed by
ANOVA result and /or visual investigation of the data, we compared (a) differences between soil layers per
polygon type and (b) differences between polygon types per soil layer category. If homogeneity of variances and
normality of model residuals were not given, log or sqrt transformations were applied. In case of no agreement
with model assumptions after transformation, we conducted nonparametric tests. Kruskal Wallis tests were used
to test the effects of polygon type and soil layer category, followed by pairwise two-sided Wilcoxon tests (function
pairwise.wilcox.test(), p.adjust='bonferroni'). To check for possible interactive effects in a comparable manner
as described for the lme models, we applied Wilcoxon tests on respectively subsetted parts of the dataset and
visually checked the distribution of the examined parameter among the soil layer categories of each polygon type.
We employed the phyloseq package (McMurdie and Holmes, 2013) for handling the multivariate datasets on
amplicon sequencing and SOM chemical composition. Following (Alteio et al., 2021), we applied a centered log-
ratio (clr) data normalization (microbiome:: transform(phyloseq.object, "clr") and calculated euclidean distance
matrices (phyloseq::distance(phyloseq.object, "euclidean"). We perfromed Principal Component Analyses



(PCAs) for visualization, employing the function 'phyloseq::ordinate()'. We used Permutational Multivariate
Analysis of Variance (PERMANOVA) to explore the effects of polygon type and soil layer category and their
possible interaction (adonis()-function implemented in vegan with 999 permutations and p-adjust.m='bonferroni';
vegan version 2.5-7, Oksanen at al., 2020). We tested differences between polygon types and/or soil layer
categories by pairwise multilevel comparisons (paiwise.adonis()-function implemented in vegan with 999
permutations and p.adjust.m='bonferroni', (Martinez Arbizu, 2020). In case of interactive effects, we used
subsetted datasets for making pairwise tests. Analogously as described for the lme model, we tested for (a)
differences between soil layers within each polygon type and (b) for differences between polygon types for each
soil layer. As PERMANOVA test results are sensitive to heterogenous dispersions among the investigated groups,
we tested their variance of dispersion using Permutation Tests for Multivariate Dispersion Homogeneity
(PERMDIST), implemented in vegan (vegan::betadisper()-function) using 999 permutations and the argument
'bias.adjust=T' for unequal sample numbers (Anderson, 2017). We used Venn diagrams
(get_vennlist(phyloseq.object) for visualizing the fraction of shared versus unique pyrolysis products and/or
microbial ASVs among polygon types and soil layers respectively (MicrobiotaProcess package, (Xu et al., 2022).
For more details see Supplementary.
**3 Results**
*3.1 Physicochemical soil parameters and stoichiometry*
We characterized soil properties either by polygon type (averaged across all soil layers) or by soil layer (averaged
across the three polygon types), respectively. LCPs differed from the other polygon types with respect to several
parameters, for example, higher soil C and N contents, and lower soil P concentrations, particularly in their organic
layer (Supplementary Table 2(a)). Their comparatively high C and N contents together with the relatively lower
P content led to soil C:P and soil N:P ratios being on average double as high as in the other two polygon types.
The active layer was also on average 10 cm deeper in LCPs than in HCPs (Supplementary Table 1(b)).
In situ temperatures decreased along the soil profile from approximately 5.6 °C at the surface to 1.4 °C at the
permafrost table (Supplementary Table 1(b)). Several other differences in physicochemical properties were
observed across the examined soil layers, with the most pronounced contrasts occurring between organic topsoils
and mineral subsoils. The organic layer for instance was characterized by an 8 times higher field water content,
and it also contained on average more than 5 times as much C and N, and more than 12 times higher dissolved
organic carbon (DOC) and total dissolved nitrogen (TDN) concentrations than the mineral layer (Supplementary
Table 2(b)). In contrast, cryoturbated material and upper permafrost soils displayed relatively similar values at an
intermediate range. Although C:N ratios were consistent across soil layers, mineral subsoils were characterized
by significantly lower C:P and N:P ratios. The ratio of DOC:TDN was nearly twice as high in the organic and
mineral layer compared to the cryoturbated and permafrost layer.

*3.2 Soil organic matter composition*
We assessed the chemical composition of organic matter pools in different polygon types and soil layer categories
using pyrolysis-GC/MS. We noted a particularly distinct fingerprint pattern of LCP soils, whereas those of FCP



and HCP soils were similar (Fig. 2(a)). Correspondingly, LCPs also shared much less pyrolysis products with the
other polygon types than were shared among FCPs and HCPs (Supplementary Fig. 2(a)). At the same time, LCP
soils also had the smallest fraction of polygon-type specific pyrolysis products. Comparing SOM compound class
abundances between polygon types revealed that LCP soils harbored significantly more lignin- derived substances
than the other polygon types in absolute and relative terms, and higher absolute abundances of aromatics &
phenols, lipids, and general & unknown compounds than FCP soils by trend (Supplementary Fig. 3(a),4(a)).
Shifts in the chemical composition of SOM also occurred between soil layers, and this effect was comparatively
stronger than the effect of polygon type (Fig. 2(b)). Organic topsoils and mineral subsoils were characterized by
rather distinct SOM pools. Their chemical fingerprints differed significantly from those of all other soil layers and
included a notable proportion of layer-specific pyrolysis products (10 % and 7 % of all considered pyrolysis
products, respectively; Supplementary Fig. 2(b)). The SOM fingerprints from the cryoturbated material and the
permafrost layer could not be distinguished from another and only contained a small fraction of unique pyrolysis
products (2.5 % of all pyrolysis products, respectively). Absolute abundances of SOM compound groups closely
reflected the underlying soil carbon concentrations (Supplementary Table 4). The highest absolute abundances
across all six SOM groups were found in the organic topsoil, followed by intermediate levels in cryoturbated and
permafrost layers, and the lowest abundances in the mineral subsoil, accordingly (Supplementary Fig. 3(b)). To
account for differences in total carbon content, it was hence more suitable to compare the relative abundances of
SOM compound classes across soil layers. In relative proportions, aromatic and phenolic compounds were for
example highest in mineral subsoils, whilst lowest in organic topsoils (Supplementary Fig. 4 (b)). Similarly, N-
containing compounds were most scarce in the mineral layer in absolute terms, while in relative terms,
cryoturbated material was the most limited.



Permanova:
Polygon: p=0.001 (F=3.46)
Soil layer: p=0.001 (F=7.42)
Polygon x Soil layer: p=0.596 (F=0.94)

(a)

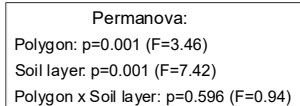

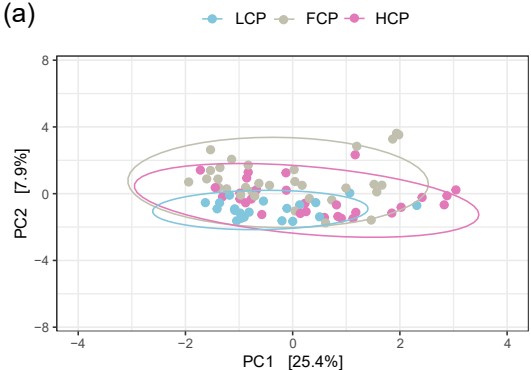

(b)

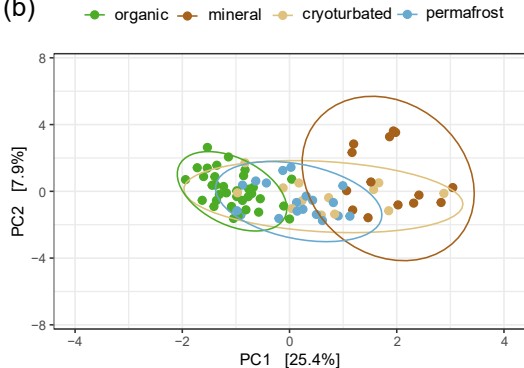


**Figure 2. Soil organic matter (SOM) composition across ice-wedge polygon types (a) and soil layer categories (b).** Principal component analysis (PCA) was performed on center-log-ratio (clr) -transformed abundances (mg C g$^{-1}$ DW) of 534 considered pyrolysis products. Permutational ANOVA (Permanova) was performed on Euclidean distance matrices, followed by pairwise adonis test for individual comparisons. Ellipses represent 95% confidence intervals.

**(a)** SOM composition differed between polygons, with a distinct fingerprint in LCP soils (LCP vs. FCP: p = 0.003, F = 3.62; LCP vs. HCP: p = 0.018, F = 2.87; FCP vs. HCP: p = 0.072, F = 2.06). (Betadisper_polygons: p=0.102, F=2.36; n_LCP=20, n_FPT=32, n_HCP=29).

**(b)** SOM composition differed across soil layers, with distinct fingerprints in organic topsoils and mineral subsoils (organic vs. mineral: p=0 006, F=17.01; organic vs. cryoturbated: p = 0.006, F = 6.43; organic vs. permafrost: p = 0.006, F = 5.47; mineral vs. cryoturbated: p = 0.012, F = 3.12; mineral vs. permafrost: p = 0.006, F = 7.04). SOM fingerprints of the cryoturbated material and permafrost layer could not be distinguished from another (p= 0.252, F = 1.64). (Betadisper_soil layers: p=0.061, F=2.57; n_organic=35; n_mineral=14; n_cryoturbated=13, n_permafrost=19).



### 3.3 Microbial Communities

We analyzed microbial community composition by sequencing the bacterial and archaeal 16S rRNA gene and
fungal ITS1 region. We used ddPCR derived gene copy numbers to estimate microbial abundances, and to
quantitatively assess specific phyla of interest.

### 3.3.1 Bacterial and archaeal abundance proxies, alpha-, and beta- diversity

The dataset comprised 41 bacterial and six archaeal phyla. Bacteroidota (28.6 %), Proteobacteria (19.8 %),
Verrumicrobiota (16.5 %), Acidobacteriota (14.2 %), and Actinobacteriota (4.5 %), represented the five most
abundant phyla, and together accounted for 84 % of all obtained ddPCR-corrected reads. Archaea, by comparison,
only comprised 1.8 % of the overall community. Taxonomic resolution was limited for a substantial proportion
of the prokaryotic community members, as approximately more than a third (1173 out of 3780) of all bacterial
and archaeal ASVs remained unclassified at the family level.
Compared to other polygon types, LCP soils exhibited lower richness, Shannon diversity, and reduced abundance
of bacteria and archaea (Supplementary Table 6 (a)). When 16S rRNA gene copy numbers were expressed per
gram of dry soil, the lower abundance in LCP soils was only visible in the organic layer (interactive effect).
However, when normalized to differences in soil carbon content, bacterial and archaeal abundance was
consistently lower across all soil layers of LCPs. The structure of bacterial and archaeal communities in LCP soils
also differed significantly from those in FCP and HCP soils (Fig. 3(a)). No significant difference was, however,
found between FCP and HCP communities. The distinctiveness of LCP communities was also reflected in other
observations. FCPs and HCPs, for example, shared 32% of the total number of detected ASVs, whereas LCPs
shared only 5% with FCPs, and 2.5% with HCPs (Supplementary Fig. 5(a)). LCP soils also had the highest
proportion of polygon-specific ASVs relative to total ASVs per polygon type (31 % for LCPs, 24 % for FCPs,
and 18 % for HCPs), despite harboring a much lower total number of bacterial and archaeal ASVs (1449 for LCPs,
2791 for FCPs, and 2471 for HCPs). When comparing absolute abundances patterns across polygon morphologies,
we found that three of the five most dominant bacterial phyla, namely Proteobacteria, Verrucomicrobiota, and
Actinobacteriota were significantly less abundant in soils of LCPs compared to FCPs and HCPs (Supplementary
Table 8). Armatimonadota, Bdellovibrionata, Cyanobacterota, and Gemmatimonadota also had reduced
abundances compared to either FCPs or HCPs, and a few phyla, such as RCP2-54, or WPS-2 were almost absent
from LCP soils. Archaea, contrastingly, were notably enriched in LCP soils, and especially in its topsoil layer.
LCP topsoils alone accounted for 65 % of all archaeal gene copies in the dataset and were characterized by a
particularly high abundance of Euryarchaeota, Crenarchaeota, Micrarchaeota, Nanoarchaeota, and Halobacterota.
Indeed, LCP communities were especially distinct in their topsoil layer, as indicated by the significant interaction
between polygon type and soil layer (Fig. 3; Supplementary Table 7). Compared to topsoils in FCPs or HCPs,
LCP topsoils showed a reduced abundance of Acidobacteriota, and Planctomycetota, but elevated abundances of
several phyla, including Desulfobacterota (sulfate reducers), Myxococcota (predators), Methylomirabilota
(methane oxidizers), and more.




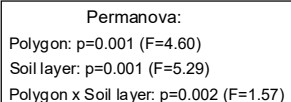

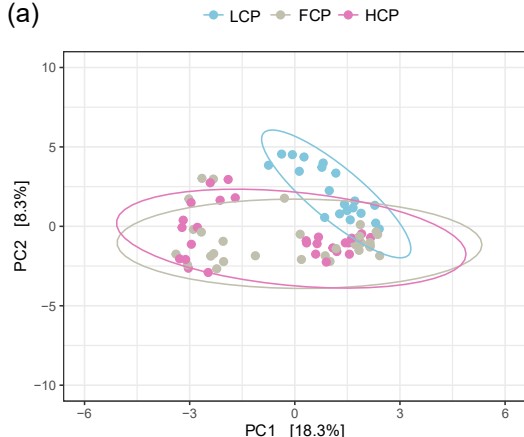

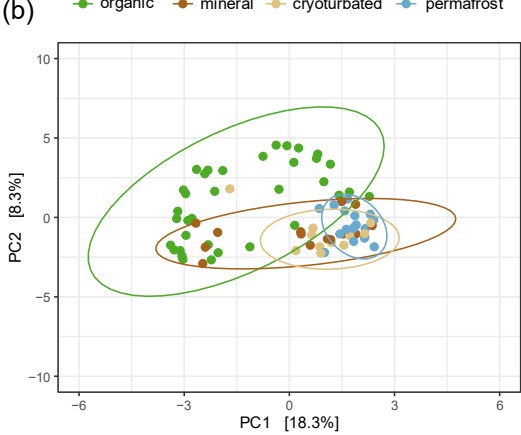


**Figure 3. Bacterial and archaeal community composition across ice-wedge polygon types (a) and soil layer categories (b).** Principal component analysis (PCA) was performed on center-log-ratio (clr) -transformed abundances (gene copy number corrected reads g⁻¹ DW) of 3780 considered bacterial and archaeal ASVs. Permutational ANOVA (Permanova) was performed on Euclidean distance matrices, followed by pairwise adonis test for individual comparisons. Ellipses represent 95% confidence intervals.

**(a)** Bacterial and archaeal community composition differed between polygons, with a distinct fingerprint in LCP soils (LCP vs. FCP: p=0.003, F=4.76; LCP vs. HCP: p=0.003, F= 5.93; FCP vs. HCP: p=0.183, F=1.56). A significant interactive effect occurred: the LCP organic and permafrost layers hosted unique communities compared to their FCP or HCP counterparts (statistical details in Supplementary Table 7). Note that inhomogeneous dispersions between polygon types may have affected these results (Betadisper_polygons: p=0.006, F=6.32; n_LCP=20, n_FPT=30, n_HCP=29).



**(b)** Bacterial and archaeal community composition differed across soil layers, with fingerprints differing between the organic and the permafrost layer in all polygon types (LCP: p=0.006, F=3.37; FCP: p=0.006, F=4.59; HCP: p=0.006, F=5.71). The fingerprints of the cryoturbated material and the mineral layer communities could not be distinguished from another (FCP: p=0.972, F=1.51; HCP: p=0.426, F=1.31). A significant interactive effect occurred: in FCPs and HCPs, topsoil and permafrost communities differed from communities of other soil layers in addition (statistical details in Supplementary Table 7). Note that inhomogeneous dispersions between soil layers may have affected these results (Betadisper_soil layers: p=0.001, F=31.72; n_organic=35, n_mineral=14, n_cryoturbated=11, n_permafrost=19).

Overall, soil layer had a stronger influence on microbial richness, alpha diversity, and abundance patterns than polygon morphology. All metrics declined significantly from the organic topsoil to the permafrost layer (Supplementary Table 6(b)). For instance, the organic layer harbored twice as many bacterial and archaeal ASVs as the permafrost layer, and Shannon diversity dropped by approximately 20 % along the same gradient. In terms of absolute abundance, organic topsoils accounted for 75 % of all 16S rRNA gene copies per gram of dry soil, compared to only 3.6 % in the mineral subsoil, 11.7 % in cryoturbated material, and 10 % in the permafrost layer. Notably, even after accounting for differences in soil carbon content, organic topsoils remained a clear microbial abundance hotspot. Regardless of polygon type, bacterial and archaeal community structure differed significantly between the organic and permafrost layers, whereas communities in cryoturbated material and adjacent mineral soils were indistinguishable (Fig. 3(b)). The communities in organic topsoils were particularly distinct, with approximately 40 % of the total number of bacterial and archaeal ASVs being unique to the layer (5 % to the mineral layer, 2 % to the cryoturbated material, and 3.5 % to the permafrost layer; Supplementary Fig. 5(b)). Likewise, approximately half (48 %) of all taxa that were found in topsoils were unique to it. The proportion of bacterial and archaeal ASVs that the organic layer shared with other layers declined with increasing soil depth (the organic layer shared 15 % with the mineral, 4 % with the cryoturbated, and 2.5 % with the permafrost layer, respectively). The comparison of absolute abundances of phyla between soil layers showed that all the above-mentioned five most abundant ones occurred in higher abundances in the organic topsoil layer compared to the permafrost layer (Supplementary Table 9). Predominantly associated with the permafrost layer were Campylobacterota, Caldisericota, Cloacimonadota, and Firmicutes, but also the fraction of unknown taxa at phylum level was notably high.

### 3.3.2 Fungal abundance proxies, alpha-, and beta- diversity

The dataset comprised seven fungal phyla, with Ascomycota and Basidiomycota being the most abundant ones. Together, they comprised approximately two thirds of all fungal ddPCR-corrected reads. More than 50 % of all fungal taxa (873 out of 1604) could not be assigned at phylum level, comprising the 'last third' of ddPCR corrected reads.

Fungal community patterns largely mirrored those observed for bacteria and archaea. Fungal richness, Shannon diversity, and abundance were lower in LCP soils compared to FCPs and HCPs (Supplementary Table 6(a)). This lower abundance was restricted to the organic layer when ITS1 gene copy numbers were expressed per gram dry soil (interactive effect) but became evident across all layers in LCPs after normalization to soil carbon content. Fungal community structure also differed significantly in LCP soils (Fig. 4(a)), with only ~5 % of taxa shared between LCPs and the other polygon types (Supplementary Fig. 6(a)). Despite harboring fewer fungal taxa overall, LCP soils contained the highest proportion of polygon-specific ASVs relative to total ASVs (LCPs 60 %, FCPs



55 %, HCPs 53 %). Fungal phylum abundance patterns reinforced the observation of a less rich, less diverse, and compositionally distinct community in LCP soils. Basidiomycota and Chytridiomycota were less abundant than in soils of FCPs and HCPs. Unknown fungi were an order of magnitude less abundant in every layer, Kickxellomycota were nearly absent, and Mortierellomycota, Rozellomycota and Zoopagomycota were absent from LCP soils (Supplementary Table 9).

Fungal richness, diversity, and abundance declined markedly with soil depth (Supplementary Table 6(b)). The organic layer contained twice as many fungal ASVs as the permafrost layer, and Shannon diversity dropped by roughly one third. In terms of abundance, 96.2 % of all ITS1 gene copies were located in the organic topsoil, while fungal abundance outside the topsoil was minimal (2.4 % in cryoturbated material, and <2 % combined in mineral and permafrost layers). Even after normalization to soil carbon content, the organic layer remained a clear hotspot for fungal marker genes. Fungal community structure in organic topsoils was distinct from that in deeper layers, regardless of polygon type (Fig. 4(b)). Nearly half of all fungal ASVs in the dataset were unique to the organic layer, and the fraction of total fungal ASVs that the organic layer shared with other layers decreased with increasing depth (Supplementary Fig. 6(b)). More than two thirds (69 %) of all taxa that occurred in the topsoil layer were also unique to it, while this was approximately one third in the other layers (31% in the mineral and cryoturbated layers, 39% in permafrost). Moreover, all detected fungal phyla were present in the organic layer and in abundances that were at least an order of magnitude higher than in the other layers (Supplementary Table 9).



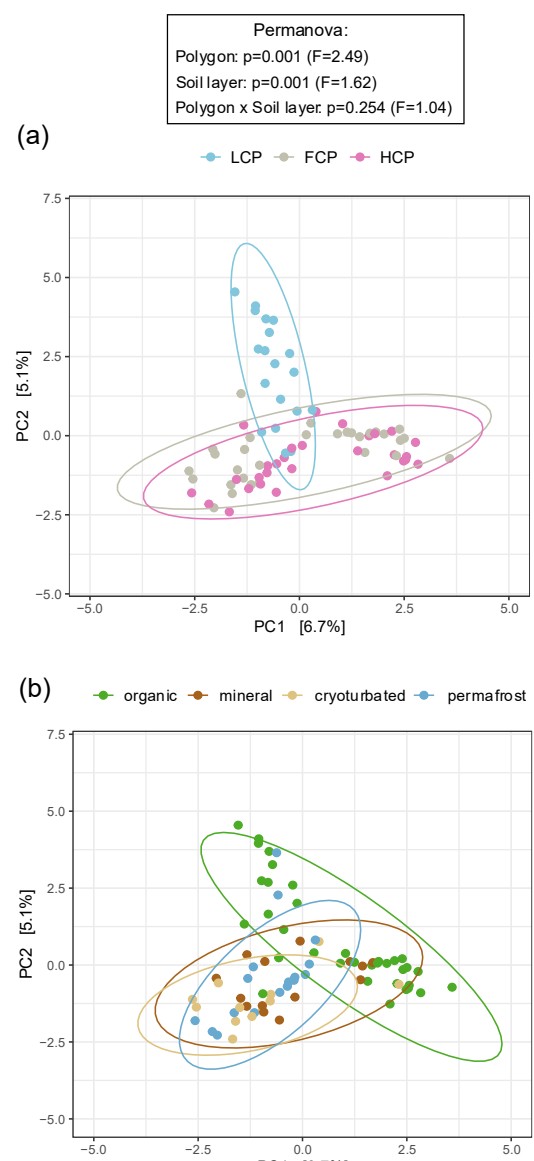

462

**Figure 4. Fungal community composition across ice-wedge polygon types (a) and soil layer categories (b).** Principal
component analysis (PCA) was performed on center-log-ratio (clr) -transformed abundances (gene copy number corrected
reads g$^{-1}$ DW) of 1604 considered fungal ASVs. Permutational ANOVA (Permanova) was performed on Euclidean distance
matrices, followed by pairwise adonis test for individual comparisons. Ellipses represent 95% confidence intervals.

**(a)** Fungal community composition differed across polygons, with a distinct fingerprint in LCP soils (LCP vs. FCP p = 0.003,
F = 2.70; LCP vs. HCP p = 0.003, F=3.13; FCP vs. HCP: p=0.009, F=1.72). Note that inhomogeneous dispersions between
polygon types may have affected these results (Betadisper_polygons: p=0.039, F=3.59; n_LCP=20, n_FPT=32, n_HCP=29).





**(b)** Fungal community composition differed between soil layers, with a distinct fingerprint in the organic layer (organic vs.
mineral p=0.024, F=1.51; organic vs. cryoturbated p=0.006, F=2.01; organic vs. permafrost: p=0.006, F=1.91). Note that
inhomogeneous dispersions between soil layer categories may have affected these results (Betadisper_soil layers: p=0.003,
F=5.83; n_organic=35, n_mineral=14, n_cryoturbated=12, n_permafrost=17).
*3.4 Potential extracellular enzymatic activity*
We expressed enzyme rates per unit of soil carbon to account for a potential effect by diverging soil C
concentrations (Supplementary Table 10, Supplementary Fig.6). Overall, potential enzymatic activity per unit soil
C varied greater between soil layers than between polygon types (Fig. 5). While activities of P- and S-cycling
enzymes did not differ consistently between polygon types, C- and N-cycling enzyme rates were lower in LCP
soils compared to FCP and HCP soils. However, this was mainly driven by differences in specific soil layers.
LCPs exhibited reduced enzyme activities, with lower rates of betaglucosidase, exoglucanase, and leucine
aminopeptidase in the organic layer, and diminished exochitinase activity in the permafrost layer (Supplementary
Table 11). Generally, enzyme activity profiles were relatively uniform across soil layers in LCPs but showed
pronounced vertical variation in other polygon types. In FCPs and HCPs, activities of betaglucosidase,
exoglucanase, and phosphatase still peaked in the organic layer despite normalization to soil carbon. In HCPs,
exochitinase activity however reached its maximum in the cryoturbated and permafrost layers, whereas no layer-
specific differences occurred in LCPs and FCPs.



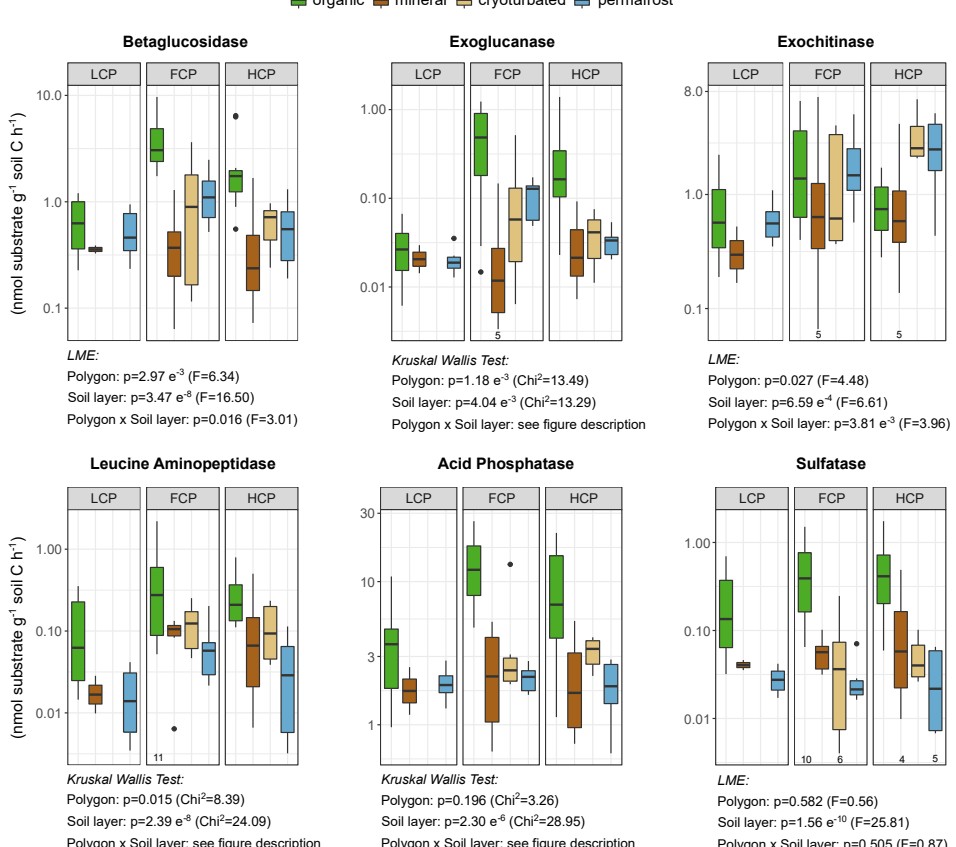

**Figure 5. Extracellular enzymatic activities in investigated soil layer categories and ice-wedge polygon types.**

Rates (nmol substrate g$^{-1}$ soil C h$^{-1}$) are depicted on a log-scale for improved readability (LCP_organic: n=12, FCP_organic: n=12, HCP_organic: n=11, LCP_mineral: n=2, FCP_mineral: n=6, HCP_mineral: n=6, FCP_cryoturbated: n=7, HCP_cryoturbated: n=6, LCP_permafrost: n=6, FCP_permafrost: n=7, HCP_permafrost: n=6; with deviations stated below respective boxplots). Effects of polygon type and soil layer category are stated under respective panels (LME ANOVA type III or Kruskal Wallis test results). LCP soils had lower rates of C- and N-cycling enzymes than FCPs and HCP soils. The effect that was largely driven by soil layer-specific differences (reduced betaglucosidase-, exoglucanase-, leucine-aminopeptidase-rates in the LCP organic layer, reduced exochitinase rates in the LCP permafrost layer). In LCPs, enzyme activities were also rather constant across soil layers, but varied considerably in FCPs or HCPs (betaglucosidase, exoglucanase, exochitinase, acid phosphatase). Statistical details in Supplementary Table 11. Please note that N-, P- and S-depolymerizing enzymes are inherently related to the C-cycle, which hinders the clear differentiation between microbial nutrient- versus C acquisition.

## 4 Discussion

We examined the concerted influences of ice-wedge polygon morphologies and soil layers on soil organic matter pools and microbial communities, aiming to identify unifying drivers that can inform permafrost carbon–climate feedback models in the spatially heterogeneous Arctic lowland tundra.



### 4.1 Effects of Polygon Morphology

Across most measured characteristics, FCP and HCP soils showed consistent similarities, while LCP soils stood out as markedly different. This disparity reflects inherent features of LCPs, including their distinctive vegetation cover, peaty soils, and persistent summer water saturation. Together, these factors shape a wide range of physical, chemical, and biological processes and likely caused the divergences in soil properties, SOM characteristics, and microbial communities that we observed in LCPs. Water logging, for example, which is known for largely restricting microbial decomposition (Dungait et al., 2012; Schädel et al., 2014), likely has contributed to the elevated soil C and N concentrations in LCP soils. Yet, for biogeochemical models, organic matter quality is considered equally important as its quantity (Jansson and Taş, 2014; Mackelprang et al., 2016; Treat et al., 2014). Although soil C:N ratios are widely used as a convenient and easily available proxy for OM availability (Malmer and Holm, 1984; Schädel et al., 2013, 2014; Weiss et al., 2016), the metric might not always provide a detailed picture. In our study, polygon types showed similar soil C:N ratios (Supplementary Table 1(b)), yet pyrolysis-GC/MS fingerprinting revealed pronounced differences in SOM quality (Fig. 2(a)). LCP soils exhibited the least variation in SOM composition along both PCA axes and contained the lowest absolute and relative shares of polygon-specific pyrolysis products (Supplementary Fig. 2(a)). Together, these observations suggest a more uniform SOM profile in LCPs, which likely reflects the relatively homogeneous nature of peat derived from graminoids and peat moss. In contrast, less uniform SOM fingerprints in FCPs and HCPs likely relate to the comparatively higher diversity in plant cover and the associated broader range of inputs via litter and exudates. Beyond vegetation imprints, SOM composition also reflects prevailing redox conditions and microbial activity (Herndon et al., 2020; Mackelprang et al., 2016). Strongly controlled by moisture and oxygen availability (Schmidt et al., 2011; Weintraub and Schimel, 2003), decomposition is generally less efficient under anaerobic than under aerobic conditions (Brune et al., 2000). Bioavailable carbohydrates and organic acids are preferentially consumed in anaerobic pathways (Tveit et al., 2013), whereas structurally more complex compounds, such as long-chained lipids and unsaturated hydrocarbons, are less accessible to microbes (Wilson et al., 2022) and may accumulate. Anaerobic environments also hinder the breakdown of lignin and phenolic substances in anaerobic layers due to reduced activity of oxidative enzymes (Tveit et al., 2013). In Sphagnum-rich peat, this effect may be amplified by moss-derived polyphenols such as sphagnan, which directly inhibit microbial activity (Fofana et al., 2022; Turetsky, 2003). Cold temperatures and waterlogging, which generally constrain decomposition in tundra soils, may have acted together with polyphenolic inhibition to contribute to the elevated abundances of lignin-derived compounds, aromatics and phenolics, and lipids in LCP soils (Supplementary Fig. 3(a)).

Polygon morphology shaped microbial abundance, diversity, and enzymatic activities. This aligns with previous studies that demonstrated distinct microbial communities and contrasting metabolic pathways across different polygon types (Taş et al., 2018; Wainwright et al., 2015). Although microbial biomass typically follows soil carbon distribution (Bastida et al., 2021; McGonigle and Turner, 2017), microbial abundance was lowest in carbon-rich LCP soils, even when bacterial, archaeal, and fungal gene copy numbers were normalized to soil carbon content (Supplementary Table 6(a)). This strongly indicates that another factor beyond soil carbon acts as overarching force in shaping the microbiome in this polygon type. Indeed, redox gradients have been suggested as the main driver of microbial communities through their influence on oxygen availability, pH, and organic matter quantity and quality (Lipson et al., 2015). Long-lasting anaerobic conditions as typical for LCP centers therefore



require organisms to have adapted their metabolism (Tveit et al., 2013). In our dataset, this may be reflected by
the highest proportion of polygon-specific taxa in LCPs alongside a minimal taxonomic overlap with the other
two polygon types (Supplementary Fig. 5(a),6(a)). Typical of anoxic systems (Lynch et al., 2023), microbial
richness and diversity were also lower in LCPs (Supplementary Table 6(a)). In addition to the scarcity of aerobic
groups, anaerobic pathways, such as fermentation, methanotrophy, or respiration via alternative electron
acceptors, generally yield less energy than aerobic respiration (Madigan et al., 2021), confining the number of
organisms that can be sustained. In this sense, the already addressed accumulation of lignin in LCP soils may also
be linked to the low abundance of obligate aerobic Basidiomycota (Supplementary Table 9), which are key agents
in lignin degradation (Zak and Kling, 2006). Fungal communities are also strongly linked to plant composition
(Chu et al., 2011; Malard and Pearce, 2018; Wallenstein et al., 2007). For instance, sedges and mosses (e.g.,
Eriophorum sp., Sphagnaceae, and Amblystegiaceae) dominate in LCPs but lack mycorrhizal associations (Chen
et al., 2020), whereas FCPs and HCPs are richer in dwarf shrubs (e.g., Betula sp., Salix sp., and Ericaceae), with
mycorrhizal associations (Lynch et al., 2018). The absence of key mycorrhizal and saprotrophic fungi in LCPs
may thus help explain their distinct fungal community (Fig. 4(a)). Similarly, the high archaeal abundance
(Supplementary Table 8) is likely a key factor for the distinct prokaryotic fingerprint in LCPs (Fig. 3(a)). Archaea
commonly comprise only a marginal fraction of Arctic microbiomes (Gittel et al., 2014; Müller et al., 2018), but
are considered major players in peaty environments (Andersen et al., 2013; Bräuer et al., 2020; Tveit et al., 2013).
There, they mediate key biogeochemical processes such as methanogenesis, methanotrophy, or ammonium
oxidation, depending on the prevailing oxygen concentrations.
Heterotrophic microbes excrete extracellular enzymes for breaking down high-molecular-weight substrates into
smaller, assimilable compounds (Lehmann and Kleber, 2015). However, microbial enzymatic activity is governed
by the balance between substrate- availability and -need (Burns et al., 2013; Moorhead et al., 2012). The lower
potential for C- and N-scavenging enzymes in LCP soils, and particularly in their organic layer (Fig. 5), likely
mirrors the detected disparities in their microbial communities, and SOM quality. First, LCPs had lower microbial
abundance and diversity and were, for example, markedly depleted in fungi (Supplementary Table 6(a)). Fungi
often are key degraders that produce a wide array of hydrolytic and oxidative enzymes (Baldrian et al., 2010;
Schneider et al., 2012). Especially in high latitude systems, mycorrhizal fungi play a central role in hydrolytic
protein breakdown (Bending and Read, 1996; Read and Perez-Moreno, 2003), and white-rot fungi, such as
Basidiomycota, oxidatively degrade lignin and humified SOM (Hatakka, 2005; Lee et al., 2012). Although we
measured only hydrolytic enzymes, the elevated lignin and phenolic contents in LCP soils (Supplementary Fig.
3(a)) suggest that oxidative enzyme activity was also suppressed, likely due to low oxygen availability (Tveit et
al., 2013). Enzyme production is energetically costly, suggesting that aerobic communities in FCPs and HCPs
may have more energy to allocate toward enzyme synthesis than the anaerobic communities in LCPs. Third,
microbial enzyme production may have been stimulated by the greater diversity of substrates in FCPs and HCPs,
while in LCPs, polyphenols could have suppressed it (Kostka et al., 2016).

*4.2 Effects of Soil Layer*

While LCP soils consistently stood out from FCP and HCP soils, soil layer effects were more nuanced, as certain
inherent peculiarities were noted for each layer. However, we advocate that many of these patterns can be largely



explained by gradients in redox conditions, SOM content, or their interplay. For instance, soil pH was lower in
organic topsoils than in the permafrost layer (Supplementary Table 2(b)), as was also reported by (Gentsch et al.,
2018)). This likely reflects contrasting dominant processes. In oxygen-limited permafrost, soil pH may be
influenced by proton-consuming microbial processes (e.g., iron-, manganese-, sulfate-, or nitrate reduction),
whereas in topsoils, pH is strongly impacted by acidifying plant inputs. Organic acids are common root exudates
(Vives-Peris et al., 2020), but also Sphagnum mosses acidify their surrounding via their metabolism and their
galacturonic acid rich biomass (Kostka et al., 2016). Again, soil C:N ratios were surprisingly stable
(Supplementary Table 2(b)), but pyrolysis-GC/MS demonstrated SOM quality changes across soil layers (Fig.
2(b)). The PCA pattern revealed a compositional shift that closely mirrored the concomitant gradient in soil carbon
content (two-sided Spearman rank order correlation: Soil C - PCA axis 1: $\rho$=-0.84, p<2.2 e-16). An increasing
degree of organic matter transformation was further suggested by SOM compound group abundance patterns,
with more plant-derived compounds in topsoils to more microbially altered ones in mineral subsoils. The organic
layer was relatively enriched in lignins, carbohydrates, and general and unknown compounds carbohydrates
(Supplementary Fig. 4(b)), likely reflecting inputs of little decomposed, labile plant detritus, and root-derived
substrates (Kuhry et al., 2020). In contrast, the mineral layer was characterized by greater proportion of less
bioavailable groups, like aromatics, phenols, and lipids. This likely reflects lower inputs of fresh organic matter
(Iversen et al., 2015), together with restricted substrate exchange with other layers due to limited soil water and
pore space, or stabilization by minerals (Dao et al., 2022; Prater et al., 2020). As result, microbial communities in
mineral subsoils may rely more heavily on OM recycling, or on metabolizing the accumulated, less bioavailable
substrates (Weintraub and Schimel, 2003; Wild et al., 2016). The permafrost SOM pool was structurally most
similar to that of organic topsoils (Fig. 2(b); Supplementary Fig. 2(b)). This similarity suggests that the upper
permafrost comprises a considerable reservoir of relatively undecomposed organic matter that could become
microbially accessible upon thaw (Gentsch et al., 2018), and aligns with field observations of substantial structural
plant residue content in the frozen material.
While SOM composition differed most strongly between the organic and mineral layers (Fig. 2(b), Supplementary
Fig. 2(b)), microbial communities showed the greatest differences between the organic and permafrost layers (Fig.
3(b),4(b), Supplementary Fig. 5(b), 6(b)). How the microbiome changes along a permafrost soil profile, has been
of interest to a plethora of studies. These, in line with our results (Supplementary Table 6(b)), reported a strong
decline in microbial biomass (Jansson and Taş, 2014; Liebner et al., 2008; Wild et al., 2016; Wilhelm et al., 2011),
richness (Lipson et al., 2015), and diversity (Frank-Fahle et al., 2014; Jansson and Taş, 2014; Liebner et al., 2008;
Müller et al., 2018; Ping et al., 1998; Taş et al., 2018) with increasing depth.
Depth-dependent shifts from more aerobic to anaerobic microbial pathways are also reported (Frank-Fahle et al.,
2014; Mackelprang et al., 2011; Müller et al., 2018). In our study, a similar depth-dependent community shift
might have been reflected in the decreasing fraction of shared taxa between organic topsoils and the subjacent
layers (Supplementary Fig. 5(b), 6(b)). Especially under frozen conditions, aerobic groups may be constrained by
the limited availability of oxygen (Mackelprang et al., 2016). Persistent subzero temperatures and elevated salinity
in unfrozen brine channels additionally select for specialist communities with stress adaptations such as, cryo-
and osmo-protectants, enhanced membrane fluidity, and modified enzyme structures (Jansson and Taş, 2014).
Halobacterota, for example, accumulate compatible solutes under osmotic stress (Pérez-Fillol and Rodriguez-
Valera, 1986). Firmicutes are known for dormancy and spore formation (Galperin, 2016), which might be





especially advantageous strategies in the transition zone between frozen and thawed conditions. Collectively,
these conditions likely explain the reduced microbial abundance and diversity in the permafrost layer
(Supplementary Table 6(b)) and may help clarify why certain phyla were particularly dominant or rare
(Supplementary Table 8). The organic layer emerged as a particular hotspot for fungi, likely driven by multiple
factors. First, most fungi are confined to aerobic conditions, (Zak and Kling, 2006). Second, in tundra, root
biomass declines rapidly with depth (Iversen et al., 2015), which causes a strong concomitant decrease in
mycorrhizal associations (Gittel et al., 2014). Third, key decomposers of plant-derived organic matter, such as
Ascomycota and Basidiomycota (Wallenstein et al., 2007) , certainly profit from the high substrate inputs (e.g.
cellulose and lignin) into the topsoil layer (Boer et al., 2005).
Overall, the organic layer emerged as a microbial hotspot in our study, but two observations suggest that redox
conditions may have exerted a stronger influence on microbial communities than SOM gradients. First, microbial
alpha and beta diversity differed markedly between the organic and permafrost layers, despite their relatively
similar SOM profiles. Second, C-rich cryoturbated material and its adjacent C-poor mineral soil hosted
communities that were not statistically distinguishable (Figs. 3(b), 4(b)).
The organic layer also emerged as a hotspot for hydrolytic degradation (Fig. 5), likely fueled by the exceptionally
high microbial abundance per unit soil C, the comparatively diverse decomposer community (Supplementary
Table 6(b)), and the availability of a strongly plant-derived substrate pool (Supplementary Fig. 3(b)). Yet, despite
harboring the lowest microbial abundance and diversity of all layers, the permafrost layer showed comparatively
high enzyme activities (Fig. 5; Supplementary Fig. 6). This suggests that hydrolytic degradation potential is
governed more by SOM properties than by microbial community structure and supports the view that permafrost
SOM may be rapidly mobilized upon thaw.
**V. Conclusions**
Improving predictions of future water, energy, and carbon fluxes in Arctic lowland tundra requires explicit
treatment of its spatial heterogeneity. Here we demonstrate that polygon-specific signals of soil organic matter
pools and microbial communities persisted across all soil layers, while layer-specific effects were consistent
across polygon type. However, our observations also highlight the distinct characteristics of two units, low-
centered polygon soils and the organic topsoil layer. Interactions between plant cover and associated organic
matter inputs (quantity and quality), alongside redox gradients, emerged as primary cross-scale drivers at both
spatial scales. Prioritizing these units and drivers in mapping and models can support spatial upscaling efforts and
forecasts of the permafrost carbon-climate feedback.
Recent studies suggest a shift from cyclic to progressive (unidirectional) ice-wedge evolution, with low-centered
polygons increasingly transitioning into high-centered ones (Fraser et al., 2018; Kartoziia, 2019; Kokelj et al.,
2014; Jorgenson et al., 2015; Kanevskiy et al., 2017; Liljedahl et al., 2016; Nitzbon et al., 2019). Because low-
centered polygons are characterized by lower microbial abundance, reduced organic matter bioavailability, and
diminished hydrolytic enzyme potential, their transformation into high-centered polygons could markedly
accelerate soil carbon losses.
In ground-ice-rich tundra, warming accelerates geomorphic changes through ice-wedge degradation, thermokarst,
and erosion (Jorgenson et al., 2022; Turetsky et al., 2020). While under stable conditions, a cyclic evolution of



ice-wedges has been suggested (Jorgenson et al., 2015; Kanevskiy et al., 2017), several observations point to a
progressive, unidirectional trajectory, in which low-centered polygons increasingly degrade into high-centered
ones (Fraser et al., 2018; Kartoziia, 2019; Kokelj et al., 2014, Liljedahl et al., 2016; Nitzbon et al., 2019). Given
the lower microbial abundance, reduced organic matter bioavailability, and diminished hydrolytic enzyme
potential in low-centered polygons, their transformation in high-centered polygons could enhance soil carbon
losses. Although drying tends to shift soil carbon emissions from $CH_4$ towards the less potent $CO_2$ (Lara et al.,
2015; Sachs et al., 2010), aerobic decomposition typically yields greater cumulative carbon losses than anaerobic
pathways (Schädel et al., 2016).
Topsoils, identified as hotspots of microbial abundance and extracellular enzyme activity, are likely to be most
affected by soil warming. Because rising temperatures are expected to accelerate microbial activity (Hutchins et
al., 2019; Karhu et al., 2014; Schuur et al., 2015), these carbon-rich horizons have a high potential to release
amplified carbon emissions. At the same time, warming also induces active layer deepening (Solomon et al., 2007;
Westerveld et al., 2023) and abrupt thaw events, which will expose previously frozen substrates to microbial
decomposition (Graham et al., 2012; Schmidt et al., 2011). As the upper permafrost harbors a large pool of
relatively undecomposed organic matter and shows considerable potential for hydrolytic degradation, substantial
additional carbon losses could occur as thaw progresses. Yet, the concurrent melting of ground ice may also
expand anoxic conditions, which would redirect decomposition toward slower anaerobic pathways in this zone
(Schädel et al., 2016).
Nevertheless, accurate projections of tundra carbon balance require integration of multiple ecosystem processes.
Vegetation dynamics such as Arctic greening (Myers-Smith et al., 2019; Phoenix and Treharne, 2022; Wolter et
al., 2016), rhizosphere priming (Friggens et al., 2025; Keuper et al., 2020; Wild et al., 2014, 2016),, or couplings
to other biogeochemical cycles (Burke et al., 2022; Keuper et al., 2012; Treat et al., 2016), may either offset or
amplify microbial feedbacks. Even so, the pronounced disparities in soil organic matter and microbial properties
observed here provide a mechanistic foundation for future spatial modeling. In particular, distinguishing low-
centered polygons from flat- and high-centered ones, and topsoils from deeper layers, offers a tractable framework
for parameterizing and scaling soil processes across the geomorphologically complex Arctic lowland tundra.
**Data availability**
The data is accessible under: https://doi.org/10.5281/zenodo.17158574.
**Competing interests**
The authors declare that they have no conflict of interest.
**Acknowledgements**
We gratefully acknowledge the dedicated logistical support provided by the team at AWI Potsdam for the Yukon
Coast expeditions in the summers of 2018 and 2019. We thank Hugues Lantuit for establishing the foundation
and framework that enabled this research, including funding, infrastructure and permitting. We thank George
Tanski for logistical help and field assistance during sample collection in 2018, Leila Jensen for briefing us with
ddPCR measurements, Alberto Canarini for help during the development of the semi-automated pyrolysis-GC/MS



fingerprinting workflow, and Petra Pjevac for the coordination of the amplicon sequencing process. We are
especially grateful to Samuel McLeod, Frank Dillon, and Peter Archie for their assistance, support, and helpful
insights in the field. We are thankful for the support received by the Yukon Territorial Government, Yukon Parks
(Herschel Island - Qikiqtaruk Territorial Park), and the Aurora Research Institute in Inuvik.

**Author contributions**

VM conducted the field and laboratory work, curated and analyzed the data, prepared visualizations, and wrote
the manuscript with input from co-authors. AR provided the primary scientific conceptualization of the study and
principal supervisor of this research. Together with AR, GH was involved in project administration and
conceptualization. GH further provided scientific guidance throughout the project and major financial support.
MF supported project administration, fieldwork logistics and contributed expertise on the research area and
manuscript writing. Fieldwork was carried out by VM, JW, WAC, LD, RL, NS, AR, and GH. CR, JH, CUM, and
MM assisted VM with laboratory work and sample analysis. VM, CR, and MM collaborated on the conceptual
development of the pyrolysis-GC/MS fingerprinting methodology. CR also played a key role in the data analysis
related to soil organic matter and microbial community composition. HS helped with amplicon sequencing,
ddPCR assays, and provided scientific input on microbial community analysis and manuscript writing. BH was
responsible for sequencing methodology and raw data processing.

**Funding**

This work is part of the Project "Nunataryuk" and has received funding under the European Union's Horizon 2020
Research and Innovation Program (grant agreement no. 773421).

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
