# Peer review of "Spatial heterogeneity of soil organic matter and microbial community composition across ice-wedge polygons and soil layers in Arctic lowland tundra"

_EGUsphere, 2025_

## Author Comment (AC1)

We thank the reviewer for the comments and provide our point-by-point responses below (in blue).

**General comments:**

Martin and colleagues explore the microbial and soil organic matter characteristics of different types of polygonal grounds (low, flat and high centered polygons) and different soil layers within each polygon type from the organic topsoil to the transient permafrost layer. This descriptive study characterizes the soil environment through a variety of methods, giving a detailed overview of chemical and microbial members of the different layers, and even some functional potential through enzymatic assays. The authors' methodology is mostly robust – although I lack the ability to judge the appropriateness of the pyrolysis procedure –, care was taken to handle the unbalanced nature of the experimental design inherent to the sampling strategy, and the detailed supplements answer many of the questions that may arise while reading the main text. All in all I appreciated reading this manuscript and look forward to reading the following publications including deeper samples.

The content of the text is well-structured, the authors do not overreach or speculate beyond what the results warrant, and the selection of literature is appropriate to support the claims. My only concern regarding the flow of text is that the conclusions seem a little disconnected from the general tone of the study, going a bit abruptly from descriptive to prospective – perhaps an additional paragraph towards the end of the discussion that would discuss the relevance of the findings for upscaling and for the permafrost carbon-climate feedback could help smoothing this transition. Not being a native English speaker I will not judge the quality of the writing, but it reads well and appears sound, though some wording could be improved (see line by line comments below).

We thank the reviewer for the thorough evaluation of our manuscript.

The manuscript already touches upon the motivation for examining spatial heterogeneity in Arctic lowland tundra and its relevance for upscaling and modelling at several points (e.g. L112–115, L119–121, L129–131, L500–502, L644–651), with a particular focus on the potential implications of climate-change-driven changes at the two spatial scales considered: (i) polygon transitions (L658–667) and (ii) active-layer warming and deepening (L668–677).

However, we agree with the reviewer that including an explicit, integrative paragraph at the end of the discussion will improve the flow from the descriptive to the more prospective elements of the manuscript and will provide a smoother transition to the conclusions. We will therefore add a paragraph that synthesizes these aspects and highlights their implications for modelling efforts and assessments of the permafrost carbon–climate feedback.

I understand the rationale for not focusing on the polygon:layer interaction, and therefore showing the results separately with different tables/PCAs for polygon type or layer, but much of the text revolves back to the interactions (e.g. some things are consistent through LCP except in the organic layer). In addition, these interactions are themselves one of the main questions the study addresses, so I think perhaps this needs to be rethought. For instance there are instances where it

is not possible for the reader to know how a variable changes across the layers of a given polygon type because the data is presented as summaries either by layer or by polygon type.

We appreciate the reviewer's careful consideration of this aspect of the manuscript. We agree that polygon × layer interactions were part of our core research questions. However, our analyses indicate that for many of the investigated variables, patterns were primarily driven by the main effects of polygon type or soil layer, rather than by their interaction.

For example, among the 13 physicochemical parameters reported (Supplementary Tables 2a,b), only soil P exhibited a statistically significant polygon × layer interaction. We therefore consider the current presentation (i.e., showing differences among polygon types and among soil layers) to be largely sufficient for conveying the dominant patterns in these parameters.

Most polygon × layer interactions were observed for hydrolytic enzyme activities. These effects are clearly reported in L477–482 and visualized in Fig. 5, which presents layer-specific activities within each polygon type and thus allows direct comparison across both dimensions. Statistical details of the interaction effects are provided in Supplementary Table 11. A mechanistic interpretation of reduced enzyme activities in the organic layer of LCPs, and their potential links to soil organic matter properties and microbial communities, is provided in L564–576.

We acknowledge that a pronounced interaction effect was observed for bacterial and archaeal community composition (Fig. 3a,b). This interaction is explicitly addressed in L386–393, including the contribution of Archaea, Acidobacteriota, Planctomycetota, Desulfobacterota, Methylomirabilota, and other taxa to the distinctness of LCP-organic topsoil communities. Regarding the graphical representation, we consider the current approach (i.e., two PCAs organized by polygon type and by soil layer adjacent to each other) to effectively depict the key interaction (i.e., LCP_organic vs. FCP_organic/HCP_organic). We explored the option of a single PCA including all polygon × layer subgroups but this resulted in substantial visual clutter and reduced interpretability.

Polygon × layer interactions were also detected for microbial abundances (Supplementary Tables 6a,b). However, after normalizing bacterial and archaeal gene copy numbers to soil C content, these interaction effects disappeared, yielding significant main effects of polygon type and soil layer, as discussed in L535–540. For fungal abundances, the interaction persisted after normalization. However, the dominant ecological pattern, namely higher abundances in organic topsoils compared to the permafrost layers, was consistent across all polygon types. We therefore consider further elaboration on layer-wise differences beyond this pattern unnecessary.

We, however, agree with the reviewer that the presentation of Supplementary Tables 6a,b could be facilitated. For this, we will add graphical representations, analogous to the enzyme plots (Fig. 5), for bacterial, archaeal, and fungal abundances, as well as for observed taxon richness and Shannon diversity, in the revised Supplementary Material.

Finally, we want to emphasize that all underlying data are publicly available via Zenodo, allowing interested readers to explore polygon × layer patterns in greater detail if desired.

**Detailed comments:**

**Introduction**

L108-111: I agree, but I would have liked to see more literature on permafrost microbial ecology in the introduction, considering it is a large part of the results of this study.

We acknowledge this comment and will strengthen the Introduction by incorporating additional background and relevant literature on permafrost microbial ecology.

The introduction could also benefit from a bit of rationale on why the authors wanted to focus on fungi and bacteria, and not microbial eukaryotes or viruses.

In our study, the term "microbial communities" is used specifically to bacteria, archaea, and fungi. We focused on these groups because they represent the primary decomposers in permafrost-affected soils and are therefore directly linked to soil organic matter turnover and biogeochemical functioning. These groups are also the most commonly characterized ones using amplicon-based approaches (16S rRNA gene and ITS regions), which is consistent with our applied community fingerprinting workflow.

We fully acknowledge the important roles of other groups, including protists and viruses. Through exerting top-down control, both hold an undoubtedly important role for soil biogeochemical cycling while remaining largely understudied. A comprehensive multi-domain assessment (bacteria, archaea, fungi, protists, and viruses), however, would have required untargeted workflows such as metagenomics or transcriptomics and was beyond the scope of the present study.

To avoid ambiguity and to clarify the terminology for readers, we will:

(i) revise the phrasing in the Introduction (L124) to "bacterial, archaeal, and fungal community structure", and
(ii) add an explanatory sentence to the Materials and Methods section (L213).

L123-127: The first two questions could be merged into the same question, as the "across scales" component is in fact addressed by using the polygon*layer design.

Thank you for the suggestion. We believe that a key strength of our study design is the simultaneous assessment of variation across two spatial scales: landscape-scale (polygon morphologies) and pedon-scale (soil layers), which is why we want to keep emphasizing this dual-scale perspective.

However, we agree with the reviewer that the original phrasing of the research questions was not optimal and will rephrase the questions to more clearly emphasize this dual-scale perspective, while distinguishing it from the assessment of independent main effects versus interaction effects within our polygon × layer design.

**Methods**

Figure 1: I would suggest to add a depiction (or schematic) of polygons and soil layers, as well as a brief graphical summary of the number of samples per polygon/layer type, to better explain the experimental design.

We agree that such schematic illustrations can be helpful for orienting readers.

During manuscript preparation, we explored options for including conceptual diagrams. However, generating accurate schematics for the different polygon types and their associated soil profiles requires substantial simplification. In addition, high-quality illustrations are already available in the literature (e.g., Liljedahl et al., 2016; Speetjens et al., 2024; Wales et al., 2020). To avoid redundancy and oversimplification refrained from introducing a respective schematic but decided to provide field photographs instead. Supplementary Fig. 1 shows examples of the three polygon types and depicts the contrasting active-layer profiles of turbic Cryosols (FCPs and HCPs) versus organic Cryosols (LCPs).

We fully agree with the reviewer that a clear summary of the sampling design is important. Accordingly, we will provide a new Table that summarizes the number of samples collected for each site and for each polygon type × soil-layer combination.

L164-165: Fix reference

Thank you for noticing this mistake.

L175-176: I find it very sad after traveling and bringing all the equipment there including a SIPRE corer to stop drilling in the transition layer. The phrasing suggests this was not the case though, so I'm looking forward to seeing the microbial and SOM data on the deeper samples from this field work, but it would have been nice to have them included here.

Thank you for this observation. Coring during the field campaigns indeed extended to a depth of approximately 1 m. The fieldwork was conducted within the framework of the EU H2020 *Nunataryuk* project, in which multiple research groups investigated different aspects of coastal permafrost systems. Deeper permafrost samples collected during the same expeditions were therefore allocated to parallel studies, including depth-resolved contaminant assessments (Lodi et al., in preparation) and spatial mapping of soil C and N distributions (Wagner et al., 2023, 2025).

The objective of our study was to focus on the soil layers most immediately affected by seasonal and future thaw and which are thus most relevant for near-term permafrost carbon mobilization. For this reason, we limited our analyses to the active layer and the transition zone. We agree that the inclusion of deeper permafrost layers would have been nice but was beyond the scope and planned objectives of this study.

L176: Here and later: "see Supplementary" - there are 55 pages of Supplementary, please refer more specifically to the relevant section, figure, table, etc.

Thank you for pointing this out. In the revised manuscript, we will specify, accordingly.

L186: Here and earlier, please clarify the cleanliness/sterility precautions taken to limit cross-contamination during sampling and homogenizing. Permafrost samples have notoriously low microbial biomass thus are sensitive to contamination.

Thank you for this comment - we agree that clarification of contamination prevention measures is important. We intend to add the following information:

Sampling was conducted under the highest level of sterility feasible under field conditions, to minimize contaminations. Nitrile gloves were worn throughout and changed between individual samples. Between samples, all tools (knives, bulk-density cylinders, and the SIPRE corer) were cleaned with water and sterilized using alcohol-based disinfectant wipes.

Permafrost cores were handled exclusively on cutting boards lined with fresh aluminum foil. The outer surface of each core was removed using a sterilized knife prior to subsampling, and material was transferred directly into sterile sample bags, which were sealed immediately. Permafrost samples were kept frozen in the field and remained unhomogenized until laboratory processing. After thawing in the laboratory, samples were homogenized by gentle mixing within the sterile bags to avoid contact with external surfaces.

Active-layer samples were root-picked and homogenized in a designated field laboratory tent on the day of sampling. Gloves were worn at all times, and all tools used during processing (e.g. tweezers, sieves, and trays) were sterilized between samples. Samples were transferred to sterile sample bags and stored at 4 °C throughout the field campaign.

L190: Please clarify what is meant with "carefully thawed", compared to simply thawed at 4°C.

Thank you for the note. The term "carefully thawed" was intended to indicate that samples were thawed at 4 °C, as higher temperatures might induce strong changes in microbial community composition. To avoid ambiguity, we will remove the word "carefully" in the revised manuscript.

L215: Please clarify here what these modifications were (I know it's in the supplements)

We will add the info accordingly.

L220: Here or in supplement, please clarify how much PhiX was added to the sequencing run.

We will clarify this point in the revised manuscript by specifying that 1 % PhiX was added to each sequencing run, following Pjevac et al., (2021).

L220 / Supplementary L117: It would be very helpful to know what range of dilution factors were necessary to dilute the extracts to 0.5 ng/µl. This could have strong impacts on inhibition between high biomass organic topsoil layers likely in the range of 50-200ng/µl and low biomass permafrost samples that may not even attain 0.5ng/µl undiluted.

We acknowledge this comment and clarify that all DNA extracts were normalized to a concentration of 0.5 ng µl$^{-1}$ prior to library preparation, ensuring comparable template input across all samples irrespective of their initial biomass. This normalization step inherently accounts for

variation in extraction yields due to differential biomass and dilutes potential PCR inhibitors in addition.

To address the specific concern raised, we note that none of the 83 DNA extracts had undiluted concentrations below 1 ng $\mu l^{-1}$, meaning that no sample required concentration or fell below the target concentration of 0.5 ng $\mu l^{-1}$.

We here provide an overview of the initial DNA concentrations and corresponding dilution factors.

| | DNA concentration (ng/µl) (mean ± stderr) | Dilution factor to reach target concentration (mean ± stderr) |
|---|---|---|
| Organic (n=35) | 22.6 ± 2.4 | 45.2 ± 4.7 |
| Mineral (n=14) | 13.7 ± 2.2 | 27.4 ±4.4 |
| Cryoturbated (n=13) | 23.7 ± 4.8 | 47.5 ± 9.6 |
| Permafrost (n=19) | 9.1 ± 2.5 | 18.1 ± 4.9 |

L229-230: Was there a particular reason not to use all of the 200-250 bp of the reads before the drop in quality for 16S, increasing the overlap to the entire target region?

Our sequencing and bioinformatic workflow followed the standardized pipeline published by Pjevac et al. (2021). The trimming parameters are optimized for highly multiplexed amplicon datasets and are part of a validated protocol that balances read quality, error rates, and merging efficiency across diverse sample types. While longer retained reads increase overlap, they may also increase error rates and compromise downstream ASV inference by extending into regions of declining quality. The parameters applied here represent a tested trade-off and were applied consistently across all samples.

L236: Although ddPCR should be less sensitive to inhibition than qPCR, was there any step taken to assess differences in inhibition across samples? I'm aware of the inhibitor removal step mentioned in supplements but that did not mention testing for inhibition afterwards.

All DNA extracts were treated with the OneStep PCR Inhibitor Removal Kit (Zymo Research, Irvine, CA, USA) to remove humic substances, polyphenols, and other common soil-derived inhibitors. For ddPCR, template concentrations were further diluted to 0.05 ng $\mu L^{-1}$ (16S rRNA gene) and 0.25 ng $\mu L^{-1}$ (ITS1 gene region), effectively reducing any remaining inhibitors to non-interfering levels (we apologize for the incorrect concentrations for ITS- samples reported in Supplementary L134).

Additionally, ddPCR provides built-in efficiency diagnostics via droplet generation and fluorescence patterns. Across all samples, we observed sufficient droplet counts and clear separation of positive and negative droplet populations, which indicates largely uninhibited reactions (inhibition would have manifested as reduced droplet numbers and inconsistent positive/negative ratios).

Given the combination of inhibitor removal, sample dilution, and the internal ddPCR controls, we refrained from additional testing for inhibition.

L246-248: I am unsure whether this was carried out on the rarefied reads, the raw reads, or the rarefied+CLR reads? Please transfer the section on the handling of contaminants and on rarefaction from Supplements to main text (Lines 144-153) as this is critical information, and clarify. In addition, 543 sequences per sample is a very low amount at which to rarefy, I would be curious as to the distribution of reads per sample, and about whether the rarefied table was created out of one random rarefaction or averaged across multiple rarefactions: a single rarefaction at a low threshold likely leads to the removal of multiple rare species for samples with high total read count and thus bias in alpha diversity estimates linked with sequencing depth – which is partly mitigated by averaging multiple rarefactions.

We thank the reviewer for highlighting these ambiguities regarding the handling of sequencing data including rarefaction and want to clarify them accordingly:

Downstream analyses were performed on amplicon datasets that had been cleaned of non-archaeal, -bacterial, or -fungal sequences, and samples with fewer than 500 obtained reads were excluded. Contaminant sequences were removed on an ASV-specific basis by subtracting the highest observed read number in one of the DNA extraction blanks from the corresponding sample reads

Rarefaction was applied to these datasets prior to calculating α-diversity, using the thresholds determined for 16S rRNA (2650 reads) and ITS1(543 reads) and the replacement-argument setting==F

We carefully evaluated the obtained read distributions, and rarefaction was performed consistently across samples. We acknowledge that rarefaction may remove low-abundance taxa in samples with higher sequencing depth. However, the method represents a controlled and widely accepted approach for ensuring comparable diversity estimates across heterogeneous sequencing datasets.

We here give an overview of the respective (raw) read numbers that were obtained from the multiplexed amplicon sequencing approach:

From the bacterial and archaeal dataset (16S rRNA gene), 2 samples were excluded from the analysis because of insufficient sequence coverage (2 x FCP_cryoturbated with only 1 and 159 obtained reads, respectively).

| 16S_reads | average | stderr | min | max |
|---|---|---|---|---|
| Organic (n=35) | 12319 | 863 | 4598 | 23779 |
| Mineral (n=14) | 5895 | 547 | 2679 | 10036 |
| Cryoturbated (n=11) | 6273 | 553 | 3749 | 8749 |
| Permafrost (n=19) | 5795 | 314 | 3764 | 8032 |

| 16S_reads | average | stderr | min | max |
|---|---|---|---|---|
| LCP (n=20) | 10540 | 1355 | 2679 | 23779 |
| FCP (n=30) | 7824 | 735 | 3696 | 18481 |
| HCP (n=29) | 8527 | 798 | 3749 | 18329 |

From the fungal dataset (ITS1 gene region), 3 samples were excluded from the analysis, because of insufficient sequence coverage (1 x FCP_cryoturbated with 99 obtained reads, 1 x FCP_permafrost with 89 obtained reads, and 1 x LCP_permafrost with 252 obtained reads).

| ITS1_reads | average | stderr | min | max |
|---|---|---|---|---|
| Organic (n=35) | 8351 | 670 | 1305 | 15015 |
| Mineral (n=14) | 7537 | 1031 | 667 | 14505 |
| Cryoturbated (n=12) | 7533 | 1998 | 2326 | 27443 |
| Permafrost (n=17) | 4019 | 512 | 899 | 7924 |

| ITS1_reads | average | stderr | min | max |
|---|---|---|---|---|
| LCP (n=19) | 6973 | 892 | 899 | 15015 |
| FCP (n=30) | 6358 | 953 | 667 | 27443 |
| HCP (n=29) | 8045 | 737 | 1305 | 14844 |

We calculated abundances of individual ASVs (gene copy number corrected reads g-1 soil DW) by multiplying the 16S rRNA gene or ITS1 copy numbers measured via ddPCR assays with their respective relative abundances from the amplicon sequencing datasets. The datasets that were used here were the relative-abundance-transformed reads after the removal of non-archaeal, -bacterial, or -fungal sequences, the exclusion of samples with less than 500 obtained reads, and the blank-correction step. Please note that rarefaction was not carried out here. Furthermore, we removed rare bacterial, archaeal, and fungal taxa (defined as containing less than 0.05 % of all gene copy number corrected reads per sample), which resulted in 3643 bacterial, 137 archaeal, and 1604 fungal ASVs to be considered in the final β-diversity analyses.

We investigated microbial community composition (β-diversity) using Principal Component Analysis (PCA). For this, abundances were center-log-ratio (clr) transformed, a widely acknowledged approach for handling compositional data (e.g., Alteio et al., 2021; Barlow et al., 2020; Gloor et al., 2017). The combination of clr-transformed, ddPCR-corrected reads ($g^{-1}$ DW) and Euclidean distances in PCA corresponds to the Aitchison distance (Aitchison, 1984).

We will make sure to clarify these points, including the distribution of reads per sample, handling of contaminants, and the rarefaction approach and thresholds in the revised manuscript.

L254: The biological relevance of ASV-level alpha diversity for fungal ITS is questionable, as the variability within this region leads to identifying individuals of the same "species" as multiple ASVs and thus inflates diversity (possibly differently across samples based on the number of individuals). A different clustering approach than DADA2, a post-DADA2 clustering of the denoised ASVs, or estimates of alpha diversity based on taxonomically resolved species/genera would all be better options here.

We agree that ITS1 exhibits substantial intra-genomic and intra-specific variability, and that fungal α-diversity estimates at the ASV level should be interpreted with caution. However, in our study, α-

diversity was used solely to describe broad patterns across polygon types and soil layers, not to infer species-level ecological processes.

Despite the limitations of ITS1, the observed patterns, namely:
(i) lower α-diversity in the wetter LCP soils compared to FCP and HCP soils, and
(ii) decreasing α-diversity with depth
are consistent with well-established ecological expectations for tundra soils and do not rely on fine-scaled sequence resolution.

Additional clustering (e.g., OTUs) might reduce absolute richness values but would likely not affect these broad ecological patterns. Because most of our downstream interpretations focus on phylum-level trends rather than species-level assignments, we consider ASV-level analyses appropriate for the descriptive nature of our study.

L254-255: This is a bit unclear to me. Does "unfiltered" refer to the removal of rare species? How about rarefaction and clr-transformation, which of those were carried out on the dataset used for alpha-diversity estimates?

Please refer to our previous response regarding the handling of sequencing data, including rarefaction, removal of rare taxa, and clr-transformation.

L258: Manganese-peroxidase or phenol oxidase would seem relevant to the study system especially considering the expected differences in water-logging, was there a particular rationale in selecting only hydrolytic and not oxidative enzymes for activity measurements? If so, please clarify

We agree that including oxidative enzyme activities, such as manganese-peroxidase or phenol oxidase, would have offered further insights into potential decomposition dynamics, particularly in systems with variable water saturation and fluctuating redox conditions. However, at the time of sample processing we prioritized generating a consistent dataset on hydrolytic extracellular enzymes due to logistical constraints. We acknowledge that this is a limitation of our study.

L264: I'm not sure where to put this so I just mention it at the mention of P values. Throughout Supplementary: please harmonize the presentation of P values. Currently some low values are shown as <0.0001, some others as < 2.2 e-16. I would suggest using <0.0001 for anything lower

We thank the reviewer for this suggestion and will harmonize the respective p-value representation.

L266: I appreciate the authors' effort in providing processed data in a usable format. For reproducibility purposes, I would recommend that in addition to the RData objects the data is made available in more interoperable and long-lasting format e.g. txt/csv/tsv. A RData object containing a phyloseq object is likely to become incompatible with some future versions of phyloseq or future major versions of R, while text files remain readable independent of software.

I would also appreciate the inclusion of data prior to processing - namely prior to rarefaction and clr-transformation.

Ideally code should also be provided, but the two points above would be a good improvement without requiring much extra work.

We thank the reviewer for highlighting important aspects of data accessibility and long-term reproducibility.

In addition to the phyloseq objects currently available on Zenodo, we will provide access to:

(i) the underlying raw data (csv-fomat), including the unprocessed ASV tables, taxonomy tables, and sample metadata.
(ii) ASV abundances (gene copy number corrected reads g-1 DW soil) prior to the exclusion of rare taxa, exported as dataframe (csv- format).

L273: I appreciate the authors' attention to the handling of their unbalanced design.

L278: Can you clarify if a table or SI table summarizes when this was the case?

We have prepared a supplementary table summarizing the applied transformations and statistical tests for all response variables. We will include it in the revised version to improve transparency.

L281: What is the rationale behind using Bonferroni correction for non-parametric posthoc tests but not in the case of parametric posthoc tests? Surely Tukey is overly conservative in both cases and could be used for the parametric EMM posthoc comparisons as well.

Thank you for this observation.

Indeed, we used Tukey's HSD adjustment for correcting p-values of pairwise comparisons of Estimated Marginal Means. This is standard for factorial mixed models and controls family-wise Type I error under the assumptions required for LMEs. Tukey HSD adjustments are well suited for a large number of pairwise comparisons.

In contrast, Tukey's HSD is not applicable to non-parametric pairwise Wilcoxon tests which follow Kruskal–Wallis analyses. For these cases, we therefore used Bonferroni correction, which provides a conservative and distribution-free adjustment of p-values for multiple comparisons. Bonferroni correction is well suited for relatively small number of pairwise comparisons as we performed them on subsetted datasets (differences between soil layers per polygon type & differences between polygon types per soil layer category).

L285-286: I do not see in Alteio 2021 that it would be recommended to apply centre-log transformation in addition to rarefaction but instead as an alternative method of normalizing for uneven read numbers. Can you clarify whether CLR was carried out after rarefaction?

This appears to be a misunderstanding. We did not intend to imply that clr-transformation was carried out on rarefied datasets or that Alteio et al. (2021) recommend combining rarefaction and clr-transformation.

To clarify, rarefaction was applied exclusively for α-diversity analyses. For β-diversity (multivariate community composition analyses), we used non-rarefied datasets that were corrected based on

ddPCR-derived gene copy numbers, filtered for rare taxa, and subsequently center-log-ratio (clr) transformed. Clr-transformation is recommended for compositional data, fully consistent with the rationale outlined in Alteio et al. (2021).

We will clarify this methods section explicitly in the revised manuscript to avoid further ambiguity.

L288-294: I am a little bit confused by the analyses carried out here, because I do not understand why a euclidean distance matrix would be created prior to PCA. If all statistical analyses were performed (betadisper, permanova, pairwise permanova) on the euclidean distance matrixes, why use PCA for visualization rather than PCoA or NMDS to ordinate the distance matrix itself? It's also more likely that the data supports the less stringent assumptions of PCoA/NMDS than PCA.

Thank you for this detailed comment and the opportunity to clarify our analytical workflow.

Our approach follows a compositional data analysis framework. For visualizing microbial community composition (β-diversity), we applied center-log-ratio (clr) transformation to the ddPCR-corrected abundance datasets and used ordination via PCA. As implemented here, PCA on clr-transformed data (performed via unconstrained RDA) is mathematically equivalent to Principal Coordinates Analysis (PCoA) with Euclidean (i.e. Aitchison) distances (Aitchison, 1984; Gloor et al., 2017). PCA was therefore chosen as eligible and interpretable ordination method for clr-transformed data.

For statistical testing, Euclidean distance matrices were calculated from the same clr-transformed datasets and used for testing multivariate dispersion (betadisper) and for Permutational ANOVA (PERMANOVA). These steps are fully consistent with the visualization approach, as both rely on the same Aitchison distance geometry.

We agree that NMDS is a commonly used alternative for working with dissimilarity matrices. However, because clr-transformed absolute abundance data are appropriately analyzed using Euclidean distances, PCA provides a straightforward and statistically coherent ordination framework (which in addition is mathematically equivalent to PCoA on Euclidean distances). In contrast, Bray–Curtis-based NMDS is primarily suited for relative abundance data and would be less appropriate for ddPCR-corrected abundance measures. During manuscript preparation, we also explored NMDS ordinations based on relative abundance data and Bray–Curtis dissimilarities. These analyses revealed patterns that were highly consistent with those obtained from PCA of clr-transformed absolute abundances. Importantly, the major conclusions regarding polygon-type and soil-layer differentiation were unaffected by the choice of ordination method.

Since we also focused on investigating broad-scale trends in phylum-level absolute abundances, we retained we retained the clr–PCA approach in the manuscript. We will explicitly clarify our workflow in the revised manuscript.

L296-299: Beyond that, homogeneous dispersion is a statistical assumption of PERMANOVA as homoscedasticity is an assumption of ANOVA. Similarly, ANOVA is robust to heteroscedasticity and non-normality of model residuals in the case of a balanced design but deviation from the

assumptions quickly becomes more problematic when the design is unbalanced. I would suggest mentioning the betadisper tests before the PERMANOVA to reflect that this is an assumption.

We acknowledge the comment and will change the order mentioning the tests in the revised Methods section, accordingly.

**Results**

L306-307: But not moisture? One would intuitively expect LCPs to be wetter, if not water-logged. Is there a reason this would not be the case, for instance sampling in September close to maximum ALT, or after particularly dry weather? When there are many parameters to present, I understand, it's a bit harder, but don't you think it's better to avoid words like "several" here and in other instances below and be more specific?

Based on the raw values, soils from LCPs did indeed tend to have higher gravimetric water contents ($2.93 \pm 0.33$ g g$^{-1}$ DW) compared to FCPs ($2.20 \pm 0.32$ g g$^{-1}$ DW) and HCPs ($2.03 \pm 0.27$ g g$^{-1}$ DW). However, this difference was not statistically significant in the linear mixed-effects model (F = 0.51, p = 0.603).

We attribute this primarily to our sampling procedure. Although LCP centers were visibly waterlogged, a substantial fraction of free water drains during coring and the removal of the peat column. This results in collected material with drier moisture contents than under the natural conditions. This artifact is also evident in Supplementary Fig. 1, where the extracted peat appears relatively drained (see panel e) despite originating from an inundated plot (as in panel a). Consequently, the reported soil water contents likely underestimate the true in situ moisture differences among polygon types. In addition, differences in organic matter content among polygon types complicate direct comparisons of gravimetric water content, as organic-rich soils inherently retain more water per unit dry mass. Therefore, water content alone provides only a partial descriptor of hydrological conditions. Nevertheless, soil organic matter pools and microbial community properties discussed in the manuscript still need to be interpreted in the context of the persistently high water table conditions in LCPs.

In addition, we agree with the reviewer regarding wording precision and will revise the mentioned paragraph (L350–354) with more specific language.

L323-324: The sentence reads like methods.

We note that the sentence represents a deliberate, brief introduction to orient readers at the start of the new subsection and was recommended by some of our co-authors for improved readability.

L350: Here and throughout, replace adonis with PERMANOVA. adonis is just the function name in vegan and pairwise.adonis is just a wrapper that creates pairwise subsets of the data, runs adonis on the subsets, then gathers the test results and applies eventual p-value adjustments

Thank you for the suggestion. We will update the revised manuscript, accordingly.

L351: Posthoc test results are hard to read in the figure legend, could they be included instead next to the ordinations? I think the betadisper results are not necessary here and could be instead in supplements

We agree that clarity of statistical results is important.

We carefully reconsidered the option of placing the post-hoc test results directly alongside the ordination panels. However, doing so introduces several practical and aesthetic issues. For the soil layer comparison, six pairwise contrasts would need to be shown. In the case of for bacterial and archaeal community composition with significant polygon × layer interactions, even more subset-specific comparisons would be required. Incorporating this volume of statistical text into the figure layout is not only impractical but also results in a visually unbalanced and overly crowded design. By contrast, reporting post-hoc results in the figure caption is more standard practice and maintains a clean and interpretable figure.

However, if the handling editor explicitly prefers the alternative layout, we will of course adapt the figure accordingly, but we believe the current approach provides the clearest presentation.

Regarding the betadisper results: we were somewhat uncertain about the reviewer's stance, as the importance of dispersion homogeneity was emphasized in an earlier comment (L296–299). We consider it appropriate to report them together with the corresponding PERMANOVA outcomes and will keep them in the figure legend for reasons of transparency and completeness.

L360-L362: Again isn't it a method rather than a result already mentioned in the concerned section?

Please refer to our answer above.

L362: I would suggest "semi-quantitatively" instead, ddPCR is not devoid of the biases inherent to all PCR approaches, and even if a PCR+ddPCR proxy is better than PCR+qPCR it remains a semi-quantitative approach.

We agree that ddPCR, like all PCR-based approaches, is subject to methodological biases (e.g. primer coverage and amplification efficiency). Our use of the term "quantitative" was therefore not intended to imply bias-free measurement, but rather to distinguish ddPCR-supported abundance estimates from relative read counts alone.

Importantly, combining amplicon sequencing with absolute gene copy numbers determined by ddPCR or qPCR is widely referred to as *quantitative microbiome profiling* and has been established as a robust framework for deriving quantitative community data (e.g. Vandeputte et al., 2017). Within this context, ddPCR provides absolute target copy numbers and substantially reduces compositional bias compared to relative abundance data.

We will clarify this definition in the Methods section by explicitly stating that "quantitative" refers to ddPCR-supported absolute abundance estimates within a quantitative microbiome profiling framework. We therefore prefer to retain the term "quantitative" rather than "semi-quantitative."

L366-367: Isn't that in part because the V4 region is not ideal for archaea, notably in terms of resolution?

We agree that the 16S rRNA V4 region generally provides lower taxonomic resolution for archaea than for bacteria, which may contribute to a lower apparent archaeal diversity and relative abundance. However, the primer set used in this study is known to capture a broad range of archaeal lineages in soil environments and has been successfully applied in previous studies.

However, even with this limitation in mind, archaea commonly represent only a minor fraction of the total prokaryotic community. The proportion we observed (1.8%) lies well within the range that is reported for Arctic and other cold-region soils (e.g., Gittel et al., 2014; Müller et al., 2018; Wilhelm et al., 2011). V4-based amplicons may underestimate archaeal diversity to some extent, but do not interfere with the descriptive ecological interpretation in our study. The major finding, (i.e., the elevated archaeal abundance in LCPs compared to other polygon types) remains robust and aligns with previous studies.

L373-374: I do not see the rationale in normalizing gene copy numbers per g SOC, particularly when there are so massive differences in SOC between the layers? I understand it for EEA, but for bacterial abundances?

We are happy to clarify our reasoning.

Because chloroform fumigation extraction was not possible in the field, ddPCR-based 16S/ITS gene copy numbers per gram of dry soil represent our most direct proxy for microbial abundance (Supplementary Table 6).

Expressing microbial gene copy numbers relative to soil C content can provide additional ecological insights, particularly in permafrost-affected soils where biomass does not necessarily scale with soil C. Even though microbial biomass commonly follows soil C contents (Bastida et al., 2021; Crowther et al., 2019; McGonigle and Turner, 2017), LCPs, for instance were characterized by high soil C and low microbial abundance. The normalization approach can therefore help illustrating that factors other than C availability, such as redox conditions constrained microbial abundance in LCPs (as discussed in L535–541).

Also, across soil layers, normalizing to soil C reveals ecologically relevant differences that are masked when abundances are expressed per gram of soil alone. For example, cryoturbated horizons often contain disproportionately low microbial biomass relative to their carbon content (Wild et al., 2016). In our case, for instance, we saw that permafrost samples spanned a relatively wide range in SOC but exhibited consistently low 16S rRNA gene copy numbers. Samples from organic topsoils covered a rather similar soil C range but showed orders of magnitude higher16 S rRNA gene copy numbers. This again suggests that microbial abundances in the permafrost layer further depend on environmental constraints such as freezing or anoxia (as discussed in L616-619).

Overall, we consider soil C normalized values as complementary to the standard per gram soil abundances. Both versions are reported transparently, and our interpretations do not rely on SOC-normalized values alone.

L381: See comment above, I won't suggest to rewrite throughout to reflect that even with corrections this remains semi-quantitative, but a compromise could be to use brackets around "absolute"

Please refer to our response to the earlier comment regarding the use this term.

L383-384: This table and Supplementary Table 9 are extremely hard to read. The presence of both uncertainty and order of magnitude, combined to the layer * polygon interaction makes it very difficult to compare numbers across groups. Notably it is not easy to say which phyla are the most abundant overall and whether this varies between polygons or soil layers. I appreciate that the authors provide these data, but think the absolute numbers would be better presented as text / CSV files in the Zenodo repository, while the data themselves would be more easily conveyed to the reader by summarizing them into a heatmap or taxa summary barplots.

We agree that Supplementary Table 8 and Supplementary Table 9 contain a large amount of information, and that the combination of mean values, uncertainty estimates, and the layer × polygon structure makes them visually dense. As noted in the table captions, it was partly not possible to perform statistical tests due to imbalanced distribution of abundances, therefore, we tried to offer informative descriptive text, accordingly.

In response to the reviewer's suggestion, we will provide heatmaps in addition to the existing tables. If the editor prefers the heatmaps over the tables, we are happy to replace them. Finally, we will also provide a CSV file with all phylum-level aggregated abundances in the Zenodo repository, so that readers can explore the dataset in alternative formats.

In response to the reviewer's suggestion, we will provide heatmaps in addition to the existing tables. Yet, the dominant phyla also visually dominate the plots, which is why we believe that the full tables remain important for transparency. If the editor feels that the heatmaps are superior to the provided tables, we are also happy to replace them. Finally, we will also provide a csv. file with all phylum aggregated abundances in the Zenodo repository, for readers to explore the dataset in alternative formats.

L386: Here I was wondering whether this may be linked to simply higher microbial biomass in LCP-organic, but the microbial biomass data is provided either as means across polygon type or means across layers, so I could not find that information.

We thank the reviewer for raising this question. We note that observed enriched archaeal abundance in LCP soils and particularly in their organic layer does not scale to differences in biomass. In fact, the LCP organic layer had significantly lower 16S rRNA gene copies $g^{-1}$ DW than the organic layer of FCPs and HCPs (as noted in the table caption of Supplementary Table 6 (a)).

What our data show instead is that Archaea are enriched both in absolute numbers and in relative contribution within LCPs:

(i) 67% of all archaeal gene copies in the dataset occurred in LCP soils, with 65% located in LCP organic layers.
(ii) The fraction of archaeal ddPCR-corrected reads relative to total 16S reads was substantially higher in LCPs (7.3%) than in FCP (1.2%) or HCP (0.2%) soils.

We agree that this distinction should be made clearer, and we will explicitly address it in the revised Discussion.

Also, Figure 3: Combining different shapes, ellipse line types and colours may help visualizing the differences due to interaction such as very distinct LCP organic topsoil communities.

We thank the reviewer for this suggestion. As noted in our response to the main comment regarding polygon × soil layer interactions, we explored visualizing all polygon x layer combinations within a single ordination. However, this approach requires 12 subgroup combinations and resulted in substantial visual clutter and reduced interpretability. We consider the present graphical layout (two separate PCAs positioned above another) effectively conveying the key interaction of interest (LCP organic vs. FCP & HCP organic) while keeping the figure readable and visually balanced.

L403: "inhomogeneous" → "heterogeneous"?

We will adapt the wording accordingly.

L414-415: This could be a true effect, but this could also be a side effect of artefacts from less successful DNA extraction in the permafrost layers, or PCR biases due to differential inhibition across layer leading to a higher sequencing depth in the organic layer. It would be useful to have the raw (before filtering/rarefaction/CLR) sequencing depth available somewhere and how it associates or not with variables such as alpha diversity metrics.

In principle, we agree that there could be technical artefacts. However, we see no indications that DNA extraction efficiency or PCR inhibition systematically biased our results. While DNA yields from the permafrost were lower (this is fully consistent with the substantially lower microbial biomass expected in frozen horizons and was also highlighted several of the reviewer's earlier comments), all DNA extracts have been normalized to the same target concentration for sequencing (please refer also to our answers about L220 / Supplementary L117).Overall, our observations are consistent with several other studies which also noted a drop in microbial biomass, richness and diversity in the permafrost layer (as discussed in L610-612).

To ensure full transparency, we will include the per-sample raw sequencing depths in the Zenodo repository together with the unfiltered ASV tables.

L427-429: Does that exceed the fact that microbial biomass in general is higher in the organic soils?

We acknowledge the comment and partly agree. Even if higher microbial biomass partly explains the higher absolute abundances of the mentioned phyla, examining absolute phylum-abundances still provides soil-layer-specific insights into the composition of the respective communities.

L429: "Supplementary Table 9" → This should be "Supplementary Table 8", correct?

Yes, thank you for noticing the mistake, we will correct it.

L430: "Caldisericota" Is this correctly assigned? Cryosericota is often observed in permafrost samples but was renamed into a new candidate phylum from Caldiserica

Thank you for pointing this out. Yes, we do think that the phylum is correctly assigned. You may check also: https://www.ncbi.nlm.nih.gov/Taxonomy/Browser/wwwtax.cgi?mode=Info&id=67814. However, we will double-check the assignment to ensure that no annotation issues occurred.

L430-431: Can you clarify what order of magnitude is meant with "notably high" here? A few % is not uncommon but more than that may suggest issues in bioinformatics processing.

Thank you for this comment. To clarify what we mean by "notably high": in our dataset, 43 % of all ddPCR-corrected reads assigned to "unknown phylum" occurred in the permafrost layer. For comparison, the corresponding proportions for the phyla that we considered enriched in permafrost were Campylobacterota (93 %), Caldisericota (81 %), Cloacimonadota (68 %), and Firmicutes (54 %). We do not interpret the high fraction of unassigned phylum-level taxa as a bioinformatical issue. The dataset was processed using a standardized and widely validated amplicon workflow (Pjevac et al., 2021), and no indications of systematic annotation problems were detected. Instead, we consider this pattern as biologically plausible and consistent with the notion that permafrost represents a reservoir of still poorly characterized taxa.

We will add the above-mentioned % in the text for better transparency.

L434-436: This is high, but in line with previous findings in permafrost systems.

We acknowledge the comment.

L446: Not a clear sentence.

We will correct it.

L450: See earlier comments on appropriateness of ASV-level estimates of fungal alpha diversity, this may simply reflect a combination of intraspecific variability and higher number of individuals

We acknowledge the comment and agree that ITS1-based ASV richness can be inflated by intra-genomic and intra-specific variability. As mentioned previously, we absolutely agree that absolute values should be interpreted with care. That said, our interpretation focuses on the directional pattern rather than the exact numerical magnitude. We consider the decrease in fungal diversity from the organic layer to the permafrost layer a robust ecological pattern that is consistent with other observations, independent of whether ASVs, OTUs, or morphological units are being used.

Figure 4: The loadings – especially of PC1 – are extremely low for fungal communities, I think another visualization than PCA may help with better depicting the differences among the communities

We assume the reviewer is referring to the relatively low proportion of variance explained by PC1 and PC2 in the fungal ordination, rather than to the loadings themselves. Indeed, the explained variance is lower than for bacteria and archaea, which in our opinion reflects the higher sequence variability of ITS1 datasets rather than a methodological limitation.

Importantly, the ecological patterns of interest, (i) clear separation of LCP fungal communities and (ii) fungal communities in organic topsoils remain well resolved in the PCA.

We also explored NMDS with Bray Curtis distance, which produced the identical ecological pattern. Nevertheless, in line with our rationale for using clr-transformed ddPCR-corrected abundance data, PCA/Aitchison-based ordination, we consider PCA as appropriate ordination approach for our dataset.

**Discussion**

L508: Yet soil moisture was the same in LCP than in the other polygons, why?

We thank the reviewer for the follow-up question and refer to our explanation in response to comment L306–307.

L527-528: Yes, but these LCPs are also covered by Eriophorum, which aerenchymae channel oxygen at depth in the soil layer. It would have been good to have some insights into root density and potential oxidative enzymatic activities.

This is a valuable point. We agree that *Eriophorum sp.* can introduce oxygen into deeper soil layers through aerenchymae and thereby influence redox conditions and decomposition processes in the immediate rhizosphere. However, a large proportion of the soil volume in LCPs would still be expected to experience suboxic or anoxic conditions. As neither root density nor oxidative enzyme activities were assessed in our study, we could only speculate about the effects in our LCP soils.

I would also suggest referring to Freeman et al 2001 when discussing phenol oxidase in peat soils

Thank you for the remark, we will certainly do so in the revised manuscript.

L541-542: Since the LCP/HCP are supposed to be cyclic, such differences would have most likely disappeared when integrating over longer periods of time, for instance by going deeper into the permafrost layer and not just in the transition layer. Using the deeper layers for this study would have been a very nice way to test this statement

This is an interesting thought. We agree that microbial communities in deeper, older permafrost primarily reflect the paleoenvironmental conditions at the time of permafrost formation (e.g., climate, vegetation) and are largely preserved because immigration of new taxa is strongly limited under frozen conditions (Ernakovich et al., 2022; Waldrop et al., 2025). Under a strictly cyclic model

of polygon evolution (Jorgenson et al., 2015; Kanevskiy et al., 2017), deeper permafrost layers may indeed show more homogenized microbial signatures between LCPs and HCPs.

However, our study was not designed to incorporate older, deeper permafrost layers and test this long-term framework. We deliberately focused on the biogeochemically active layers (organic topsoil, mineral subsoil, cryoturbated material, and the permafrost transition layer), that are most impacted by aboveground and belowground differences among polygon types and by future climate change.

L584: Presumably other soil layers than the permafrost should also be oxygen-limited, at least in the LCP no? Would it be possible to get an indication of the typical water table depth in the different polygons?

We thank the reviewer for this comment. We agree that oxygen limitation is not confined to the permafrost layer and can occur throughout the active layer, particularly in LCPs where water tables are high. Based on field observations, LCP centers were fully inundated in Ptarmigan Bay and Whale Bay. The Komakuk Beach Area seemed a little drier overall and the water table in LCPs typically reached ~10-20 cm below the surface. FCPs and HCPs by comparison had ~ 30–50% volumetric water content.

Nevertheless, we here wanted to highlight that the dominant controls on soil pH might differ with soil depth, even if both potentially experience sub/anoxic conditions. Organic topsoils are more strongly influenced by acidifying plant-derived inputs (e.g., root exudates, litter, Sphagnum metabolites). By comparison, plant influence is strongly diminished in deeper, mineral and permafrost layers, but may proportionally be more influenced by inorganic redox reactions and proton-consuming microbial processes (Fe, Mn, sulfate, or nitrate reduction).

L591: Nothing wrong with this, but why not include the non-pyrolysis soil chemistry variables e.g. SOC, pH, moisture etc. into the input variables of the PCA and then refer to their loading on PC1 instead?

Thank you for this suggestion. Indeed, including non-pyrolysis edaphic variables in the PCA would allow inspection of their loadings on PC1. Similarly, a constrained ordination (e.g., RDA or CCA) would address how much variation in SOM composition would be explained by environmental factors. However, our goal was to produce an unconstrained, composition-only ordination of SOM profiles, which displays intrinsic variation across polygon types and soil layers independent of environmental covariates. For these reasons, we retained the correlation approach mentioned in L591.

L596-597: Regardless of polygon type? One would expect high phenolic content in sphagnum peat, no?

Indeed, Sphagnum-dominated peat typically contains high absolute concentrations of phenolic compounds. This is also likely reflected in our data with high absolute abundances (mg C g$^{-1}$ soil) of aromatics and phenolics in LCP soils and organic topsoils across polygon types, where direct

sphagnum impacts prevail. Yet, when expressed as proportions (%), the mineral layer exhibited the highest relative share of aromatics and phenols, which we interpret as compositional shifts driven by preferential decomposition of comparatively labile plant-derived compounds (e.g., carbohydrates or proteins).

Notably, these patterns were consistent across polygon types, with no detected polygon x soil layer interactions in either absolute or relative SOM composition (Suppl. Figs. 3–4).

L602-605: Though not so much nitrogen in this case, contrary to suggested in Keuper et al 2012 where it was explicitly suggested to accumulate in the transition layer. Would the authors care to speculate about this discrepancy?

The reported nitrogen accumulation at the transition layer in Keuper et al. (2012) likely reflects local hydrological and biogeochemical conditions in the examined subarctic peatland. In our polygonal tundra sites, total dissolved nitrogen (TDN) concentrations in the upper permafrost were consistently high (second only to the organic topsoil), but did not show the same pronounced transition-layer accumulation. So, the differences between our findings and Keuper et al. may reflect contrasting ecosystem types (peatlands in northern Sweden versus polygonal tundra in the Canadian lowlands) and associated differences in permafrost structure, soil hydrology, and organic matter composition.

L626: One may argue that the 50% of fungi not identified to phylum level in the permafrost samples have some grounds to dispute this assumption

We thank the reviewer for this comment. The more than 50 % of fungal taxa unassigned at the phylum level refers to the overall dataset, not specifically to the permafrost samples (L434). Our conclusion that the organic layer represents a fungal hotspot is based on the distribution of fungal abundances across soil layers. ddPCR derived gene copy numbers were on average 1 order of magnitude higher in organic topsoils than in deeper layers (Supplementary Table 6(b)). Furthermore, all major identified fungal phyla showed their highest absolute abundances in the topsoil, supporting this interpretation (Supplementary Table 9).

L627-628: Yes, but this is also challenged, see for instance Hewitt et al 2020 doi:10.1111/nph.16235 on deep scavenging of permafrost-table N by mycorrhiza

We thank the reviewer for highlighting Hewitt et al. (2020) who showed that mycorrhizal fungi can be active at the thaw front and facilitate N uptake from permafrost. However, while potentially present and functional at depth, mycorrhizal associations certainly decrease in abundance in deeper soil layers. In our dataset, overall fungal abundance declined with depth, and also the relative contribution of ectomycorrhizal fungi decreased from 8% in the organic layer to 5% in the permafrost. These numbers are based on the funguild analysis that was suggested by Reviewer 2 and which we plan to incorporate into the revised manuscript.

**Conclusions**

L650-651: I'm not totally on board with that conclusion. For one, plant cover is loosely considered here, with only a brief mention of dominant vegetation in SI Table 1, after which it is not distinguished from other factors underlying the presence and different types of polygons. As the authors well know, Sphagnum, Carex and Eriophorum co-occur in the Arctic across a variety of systems that are not polygons. Then I fail to see in the manuscript what would warrant focusing on plant cover or redox rather than on microbial communities, the direction of the suggested causal link does not seem very strongly supported. Does the paper really need this statement?

We agree that plant cover and redox conditions are not isolated or experimentally constrained drivers in this study. We did not intend to imply experimental causality, but to interpret the observed polygon- and layer-specific patterns as emerging from gradients in organic matter inputs and redox regimes, which are themselves shaped in a dynamic interplay of vegetation, hydrology, microtopography, and more. We also agree that the characteristic microbial communities are certainly integral to this framework and mediate polygon- and soil-layer-specific biogeochemical processes. Yet, their abundance, community structure, and functioning are largely responses to these overarching environmental influences.

To clarify our interpretation, we will revise the paragraph to

(i) explicitly frame redox conditions and organic matter inputs as environmental gradients that are associated with microtopography, vegetation, and hydrology, rather than as isolated drivers.
(ii) clarify that these conditions shape the associated microbial communities, which mediate polygon-and soil-layer characteristic biogeochemical processes.
(iii) state in L650 that upscaling efforts in this ecosystem would benefit from accounting for polygon-type and soil-layer specificities.

L652-657: This paragraph and the next are largely duplicates.

We agree with the reviewer and suppose this was a mistake that happened during the late stage of manuscript revision. We will eliminate duplication in the revised Conclusions section.

L656-657 / 664-665: Assuming ice-wedge evolution indeed follows a unidirectional shift, do we have evidence that this shift is accelerating? Otherwise we may not witness accelerated soil carbon losses and we've been observing these losses and attributing them to FCP/HCP normal functioning until now. Continuous permafrost in a region with MAAT around -10 does not strike me as on the verge of immediate thawing, although the data mentioned are rather old and MAAT may be a couple of degrees higher nowadays. That being said I'm not familiar enough with the literature on ice wedge cycling to assess what triggers the shift from LCP to HCP and whether it is now happening faster than it has been in the last decades.

The models by Jorgenson et al. (2015) and Kanevskiy et al. (2017) describing ice-wedge polygon evolution as cyclic are conceptual and suggest stable conditions. Nowadays, permafrost soils are experiencing severe and rapid warming (IPCC, 2013, 2022). Over the last decades, widespread icewedge degradation has been documented across Arctic regions, particularly in areas of cold continuous permafrost (Fraser et al., 2018; Jorgenson et al., 2006, 2015; Liljedahl et al., 2016). This degradation is largely driven by increasing active layer thickness due to warmer and wetter winters and summers, thermokarst, and disturbances such as flooding (Kanevskiy et al., 2017; Liljedahl et al., 2016; Parmentier et al., 2024; Westerveld et al., 2023), and is therefore expected to accelerate. For example, Jorgenson et al., (2022) report that ice-wedge degradation in NE Alaska increased from 2% in 1950 to 19% in 2018, an area close to our study site. Liljedahl et al. (2016) documented polygon transitions from low-centered polygons (LCPs) to high-centered polygons (HCPs) and postulate that these transformations will expand with continued warming, with ecosystem effects through changed topography, hydrology, vegetation, and soils (Nitzbon et al., 2019). Finally, even if ice-wedge evolution is cyclic, their degradation rate will likely outpace their recovery rate given drastic future permafrost warming.

Based on these lines of evidence, we interpret the LCP → HCP trajectory as a climate-change driven process that is accelerating.

L682: "microbial feedbacks": Considering the focus of the manuscript on microbial community composition, I am missing some references to the modalities of microbial community assembly and coalescence and how this may or may not affect microbial processes upon thaw. I'm thinking of the work on community-functioning relationships that took off over the last 7 years e.g. Knoblauch 2018, Monteux 2020, Doherty 2020, Marushchak 2021, Doherty 2025, Starr 2025

We appreciate the reviewer's suggestion and now briefly acknowledge recent work on microbial community assembly and coalescence as an additional, yet only recently addressed source of complexity and variability in biogeochemical models.

L683-685: So, is there still any sense in distinguishing FCP and HCP based on the extensive assessment produced here? I'm missing a little bit of this, and perhaps something similar for the soil layers: say if I come to the Arctic as a microbial ecologist, do I need to worry much about whether I'm sampling cryoturbated or transient permafrost layers, or is it enough if I just focus on the organic/mineral distinction?

Our intention is not to suggest that FCPs and HCPs should be treated equivalently, or that deeper soil layers are less important ecological units than topsoils. In our dataset, LCPs and organic topsoils emerged as units with particularly pronounced variability in SOM and microbial properties, which makes them tractable for initial modeling and spatial upscaling efforts. Even if other polygon types and soil layers showed less pronounced characteristics, they unequivocally remain ecologically relevant. In our dataset, the organic/mineral distinction captured major SOM patterns, and the organic/permafrost distinction captured major microbial patterns. However, our observations focus on Arctic lowland tundra, and upland permafrost soils or subarctic peatlands may require consideration of different ecological units and patterns. The decision of which units to prioritize also largely depends on the specific research question, so in some studies, FCPs or cryoturbated material may warrant focused sampling.

We will rephrase L683-685 to specify this main take-away of our study.

**Supplements:**

SI Table 3: the formatting could be optimized so it is not 8 pages long

If preferred, we can also move the full table to the Zenodo repository and provide it in CSV format.

SI Table 8, 9 and 11: Please include test statistics and not just P values, as is the case in other tables. Please also include test statistics when the P values are above 0.05.

We will revise the tables.

SI Table 8,9: In the "Soil layer" and "Interactive effects" columns the post-hoc tests are replaced by less informative text, please keep the presentation consistent across columns and tables 8-9

The presentation in Tables 8 and 9 is intentional. Statistical tests are only shown for results that are discussed in the main text. And descriptive text has been used if abundance patterns were too imbalanced for statistical testing. Both is explicitly stated in the table caption. We do not plan to further revise these tables besides adding the tests statistics, as wished above. We will, however, provide the heatmaps and the corresponding raw data files in the Zenodo repository, enabling different visualization and specific re-analysis by interested readers.

**References:**

Aitchison, J.: The statistical analysis of geochemical compositions, Journal of the International Association for Mathematical Geology, 16, 531–564, https://doi.org/10.1007/BF01029316, 1984.

Alteio, L. V., Séneca, J., Canarini, A., Angel, R., Jansa, J., Guseva, K., Kaiser, C., Richter, A., and Schmidt, H.: A critical perspective on interpreting amplicon sequencing data in soil ecological research, Soil Biology and Biochemistry, 160, https://doi.org/10.1016/j.soilbio.2021.108357, 2021.

Barlow, J. T., Bogatyrev, S. R., and Ismagilov, R. F.: A quantitative sequencing framework for absolute abundance measurements of mucosal and lumenal microbial communities, Nat Commun, 11, 2590, https://doi.org/10.1038/s41467-020-16224-6, 2020.

Bastida, F., Eldridge, D. J., García, C., Kenny Png, G., Bardgett, R. D., and Delgado-Baquerizo, M.: Soil microbial diversity–biomass relationships are driven by soil carbon content across global biomes, The ISME Journal, 15, 2081–2091, https://doi.org/10.1038/s41396-021-00906-0, 2021.

Crowther, T. W., van den Hoogen, J., Wan, J., Mayes, M. A., Keiser, A. D., Mo, L., Averill, C., and Maynard, D. S.: The global soil community and its influence on biogeochemistry, Science, 365, eaav0550, https://doi.org/10.1126/science.aav0550, 2019.

Ernakovich, J. G., Barbato, R. A., Rich, V. I., Schädel, C., Hewitt, R. E., Doherty, S. J., Whalen, E. D., Abbott, B. W., Barta, J., Biasi, C., Chabot, C. L., Hultman, J., Knoblauch, C., Mackelprang, R., Onstott, T. C., Richter, A., Vishnivetskaya, T. A., Waldrop, M. P., and Winkel, M.: Microbiome assembly in thawing permafrost and its feedbacks to climate, 1–20, https://doi.org/10.1111/gcb.16231, 2022.

Fraser, R. H., Kokelj, S. V., Lantz, T. C., McFarlane-Winchester, M., Olthof, I., and Lacelle, D.: Climate Sensitivity of High Arctic Permafrost Terrain Demonstrated by Widespread Ice-Wedge Thermokarst on Banks Island, Remote Sensing, 10, https://doi.org/10.3390/rs10060954, 2018.

Gittel, A., Bárta, J., Kohoutová, I., Mikutta, R., Owens, S., Gilbert, J., Schnecker, J., Wild, B., Hannisdal, B., Maerz, J., Lashchinskiy, N., Čapek, P., Šantrůčková, H., Gentsch, N., Shibistova, O., Guggenberger, G., Richter, A., Torsvik, V. L., Schleper, C., and Urich, T.: Distinct microbial communities associated with buried soils in the siberian tundra, ISME Journal, 8, 841–853, https://doi.org/10.1038/ismej.2013.219, 2014.

Gloor, G. B., Macklaim, J. M., Pawlowsky-Glahn, V., and Egozcue, J. J.: Microbiome Datasets Are Compositional: And This Is Not Optional, Front. Microbiol., 8, 2224, https://doi.org/10.3389/fmicb.2017.02224, 2017.

IPCC: Climate Change 2013: The Physical Science Basis. Contribution of Working Group I to the Fifth Assessment Report of the Intergovernmental Panel on Climate Change [Stocker, T.F., D. Qin, G.-K. Plattner, M. Tignor, S.K. Allen, J. Boschung, A. Nauels, Y. Xia, Cambridge University Press, V. Bex and P.M. Midgley (eds.)]. Cambridge University Press, Cambridge, United Kingdom and New York, NY, USA, 1535 pp., https://doi.org/10.1017/CBO9781107415324, 2013.

IPCC (Ed.): Polar Regions, in: The Ocean and Cryosphere in a Changing Climate: Special Report of the Intergovernmental Panel on Climate Change, Cambridge University Press, Cambridge, 203–320, https://doi.org/10.1017/9781009157964.005, 2022.

Jorgenson, M. T., Shur, Y. L., and Pullman, E. R.: Abrupt increase in permafrost degradation in Arctic Alaska, Geophysical Research Letters, 33, https://doi.org/10.1029/2005GL024960, 2006.

Jorgenson, M. T., Kanevskiy, M., Shur, Y., Moskalenko, N., Brown, D. R. N., Wickland, K., Striegl, R., and Koch, J.: Role of ground ice dynamics and ecological feedbacks in recent ice wedge degradation and stabilization, Journal of Geophysical Research: Earth Surface, 120, 2280–2297, https://doi.org/10.1002/2015JF003602, 2015.

Jorgenson, M. T., Kanevskiy, M. Z., Jorgenson, J. C., Liljedahl, A., Shur, Y., Epstein, H., Kent, K., Griffin, C. G., Daanen, R., Boldenow, M., Orndahl, K., Witharana, C., and Jones, B. M.: Rapid transformation of tundra ecosystems from ice-wedge degradation, Global and Planetary Change, 216, 103921, https://doi.org/10.1016/j.gloplacha.2022.103921, 2022.

Kanevskiy, M., Shur, Y., Jorgenson, T., Brown, D. R. N., Moskalenko, N., Brown, J., Walker, D. A., Raynolds, M. K., and Buchhorn, M.: Degradation and stabilization of ice wedges: Implications for assessing risk of thermokarst in northern Alaska, Geomorphology, 297, 20–42, https://doi.org/10.1016/j.geomorph.2017.09.001, 2017.

Liljedahl, A. K., Boike, J., Daanen, R. P., Fedorov, A. N., Frost, G. V., Grosse, G., Hinzman, L. D., Iijma, Y., Jorgenson, J. C., Matveyeva, N., Necsoiu, M., Raynolds, M. K., Romanovsky, V. E., Schulla, J., Tape, K. D., Walker, D. A., Wilson, C. J., Yabuki, H., and Zona, D.: Pan-Arctic ice-wedge degradation in warming permafrost and its influence on tundra hydrology, Nature Geoscience, 9, 312–318, https://doi.org/10.1038/ngeo2674, 2016.

McGonigle, T. P. and Turner, W. G.: Grasslands and Croplands Have Different Microbial Biomass Carbon Levels per Unit of Soil Organic Carbon, Agriculture, 7, https://doi.org/10.3390/agriculture7070057, 2017.

Müller, O., Bang-Andreasen, T., White, R. A., Elberling, B., Taş, N., Kneafsey, T., Jansson, J. K., and Øvreås, L.: Disentangling the complexity of permafrost soil by using high resolution profiling of microbial community composition, key functions and respiration rates, Environmental Microbiology, 20, 4328–4342, https://doi.org/10.1111/1462-2920.14348, 2018.

Nitzbon, J., Langer, M., Westermann, S., Martin, L., Aas, K. S., and Boike, J.: Pathways of ice-wedge degradation in polygonal tundra under different hydrological conditions, Cryosphere, 13, 1089–1123, https://doi.org/10.5194/tc-13-1089-2019lc, 2019.

Parmentier, F.-J. W., Nilsen, L., Tømmervik, H., Meisel, O. H., Bröder, L., Vonk, J. E., Westermann, S., Semenchuk, P. R., and Cooper, E. J.: Rapid Ice-Wedge Collapse and Permafrost Carbon Loss Triggered by Increased Snow Depth and Surface Runoff, Geophysical Research Letters, 51, e2023GL108020, https://doi.org/10.1029/2023GL108020, 2024.

Pjevac, P., Hausmann, B., Schwarz, J., Kohl, G., Herbold, C. W., Loy, A., and Berry, D.: An Economical and Flexible Dual Barcoding, Two-Step PCR Approach for Highly Multiplexed Amplicon

Sequencing, Frontiers in Microbiology, Volume 12-2021, https://doi.org/10.3389/fmicb.2021.669776, 2021.

Speetjens, N. J., Berghuijs, W. R., Wagner, J., and Vonk, J. E.: Degradation of ice-wedge polygons leads to increased fluxes of water and DOC, Science of The Total Environment, 920, 170931, https://doi.org/10.1016/j.scitotenv.2024.170931, 2024.

Vandeputte, D., Kathagen, G., D'hoe, K., Vieira-Silva, S., Valles-Colomer, M., Sabino, J., Wang, J., Tito, R. Y., De Commer, L., Darzi, Y., Vermeire, S., Falony, G., and Raes, J.: Quantitative microbiome profiling links gut community variation to microbial load, Nature, 551, 507–511, https://doi.org/10.1038/nature24460, 2017.

Wagner, J., Martin, V., Speetjens, N. J., A'Campo, W., Durstewitz, L., Lodi, R., Fritz, M., Tanski, G., Vonk, J. E., Richter, A., Bartsch, A., Lantuit, H., and Hugelius, G.: High resolution mapping shows differences in soil carbon and nitrogen stocks in areas of varying landscape history in Canadian lowland tundra, Geoderma, 438, 116652, https://doi.org/10.1016/j.geoderma.2023.116652, 2023.

Wagner, J., Wolter, J., Ramage, J., Martin, V., Richter, A., Speetjens, N. J., Vonk, J. E., Lodi, R., Bartsch, A., Fritz, M., Lantuit, H., and Hugelius, G.: Regional synthesis and mapping of soil organic carbon and nitrogen stocks at the Canadian Beaufort coast, EGUsphere, 2025, 1–29, https://doi.org/10.5194/egusphere-2025-1052, 2025.

Waldrop, M. P., Ernakovich, J. G., Vishnivetskaya, T. A., Schaefer, S. R., Mackelprang, R., Barta, J., O'Brien, J. M., Winkel, M., Barbato, R. A., Heffernan, L., Leewis, M. C., Hewitt, R. E., Hultman, J., Sun, Y., Biasi, C., Bradley, J. A., Liebner, S., Ricketts, M. P., Muscarella, M. E., Schütte, U., Abuah, F., Whalen, E., Timling, I., Voigt, C., Taş, N., Lloyd, K. G., Siljanen, H. M. P., Rivkina, E. M., Voříšková, J., Tao, J., Liang, R., Li, Z., Lennon, J. T., and Onstott, T. C.: Microbial Ecology of Permafrost Soils: Populations, Processes, and Perspectives, Permafrost and Periglacial Processes, 1–14, https://doi.org/10.1002/ppp.2264, 2025.

Wales, N. A., Gomez-Velez, J. D., Newman, B. D., Wilson, C. J., Dafflon, B., Kneafsey, T. J., Soom, F., and Wullschleger, S. D.: Understanding the relative importance of vertical and horizontal flow in ice-wedge polygons, Hydrology and Earth System Sciences, 24, 1109–1129, https://doi.org/10.5194/hess-24-1109-2020, 2020.

Westerveld, L., Kurvits, T., Schoolmeester, T., Eckhoff, T., Overduin, P., Fritz, M., Alfthan, B., Sinisalo, A., and Mulelid, O.: Arctic Permafrost Atlas, https://doi.org/10.61523/KPJI4549, 2023.

Wild, B., Gentsch, N., Capek, P., Diáková, K., Alves, R. J. E., Bárta, J., Gittel, A., Hugelius, G., Knoltsch, A., Kuhry, P., Lashchinskiy, N., Mikutta, R., Palmtag, J., Schleper, C., Schnecker, J., Shibistova, O., Takriti, M., Torsvik, V. L., Urich, T., Watzka, M., Šantrůcková, H., Guggenberger, G., and Richter, A.: Plant-derived compounds stimulate the decomposition of organic matter in arctic permafrost soils, Scientific Reports, 6, 1–11, https://doi.org/10.1038/srep25607, 2016.

Wilhelm, R. C., Niederberger, T. D., Greer, C., and Whyte, L. G.: Microbial diversity of active layer and permafrost in an acidic wetland from the Canadian high arctic, Canadian Journal of Microbiology, 57, 303–315, https://doi.org/10.1139/w11-004, 2011.

---

## Author Comment (AC2)

We thank the reviewer for the comments and provide our point-by-point responses below (in blue).

The authors compare biogeochemical and microbial community properties among ice-wedge polygons in Arctic tundra and identify disparate compositions in saturated, low-centered polygons and organic horizons across all topographic features. Their contemporary analysis of SOM quality (from pyrolysis GC-MS), microbial community composition, and hydrolytic soil enzyme activities provides an integrated description of SOM turnover potential that both expand on previous reports from individual polygonal features and provide a self-consistent dataset for future modeling. I offer some suggestions for structural improvements that may help focus this manuscript.

**Major comments**

The Introduction section conscientiously reviews almost 75 years of permafrost soil science, but it could be better focused on a hypothesis or specific, mechanistic research question. The final paragraph of this section (lines 122-131) includes a list of questions about biogeochemical correlations across polygons and soil layers but does not describe a strategy to demonstrate causal relationships that would support the desired mechanistic insights. Is there an intrinsic model of the biogeochemical processes motivating this research that could be explicitly described and tested here to enhance the impact of this impressive work?

We thank the reviewer for this constructive comment and agree that the Introduction can be strengthened by more explicitly articulating the conceptual framework that motivates the study. While our approach is observational, it is guided by a mechanistic framework in which polygon morphology and soil layers represent the two dominant axes of environmental organization in ice-wedge polygon tundra that impose recurring constraints on edaphic properties and biogeochemical processes.

Polygon microtopography structures lateral gradients in surface hydrology, organic matter quality (via differential vegetation cover) and soil types, while vertical gradients along the soil profile impose additional physicochemical controls through changes in temperature, texture, pH, oxygen availability, and plant-derived organic matter inputs. Together, these gradients shape soil organic matter properties and microbial community characteristics across scales, including microbial abundance, diversity, and community structure, and constrain pathways of organic matter transformation and decomposition.

To address the reviewer's suggestion, we will revise the final part of the Introduction to make this conceptual framework explicit. First, we will clarify that jointly analyzing terrain-scale (polygon morphology) and pedon-scale (soil layers) variability allows us to test whether these two spatial dimensions exert predominantly independent or interacting controls.

Second, we will clarify that the mechanistic insight sought in this study concerns how cross-scale spatial organization generates distinct biogeochemical environments via differential hydrological regime, redox conditions, and organic matter inputs. These environments, in turn, shape microbial communities and constrain soil organic matter processing. Identifying these environmental linkages allows us to infer how spatial heterogeneity directs biogeochemical functioning in Arctic

lowland tundra and may inform scalable representations of its heterogeneity in ecosystem and land-surface models.

We believe that making this framework explicit will better motivate the study, clarify its mechanistic underpinnings, and strengthen the link between our analyses and their relevance for ecosystem-scale representation and modeling.

Section 3.1 & Figure 2. What factors from the pyrolysis GC-MS analyses contributed to the variation among SOM contents in LCP and organic soil samples? Were specific features heavily weighted in the principal components that can be identified to gain insight into the composition? Are these the same factors identified in LME analysis (Figures S3 & S4)?

We thank the reviewer for this question and the opportunity to clarify our analytical approach. First, we would like to emphasize that Figure 2 illustrates differences in chemical structure of SOM, rather than differences in SOM content. The fingerprints are based on the clr-normalized abundance of 534 pyrolysis-GC/MS products ("peaks") and thus reflect compositional information rather than SOM quantity.

We agree that inspection of PCA loadings can provide insight into which pyrolysis products contributed most strongly to the observed ordination patterns. To address the reviewer's request, we therefore examined the PCA loadings in detail. Separation among polygon types (Figure 2a) occurred primarily along PC2, with high-loading peaks including, for example, phenolic and N-containing compounds (e.g., phenol, 1H-imidazole, 1H-pyrrole-3-carbonitrile) as well as long-chain aliphatic compounds (e.g., 1-hexadecanol, 2-nonadecanone), and compounds of unknown origin (referred to as "Peaks 184, 1600, 1678, and 1720"). Differentiation among soil layers (Figure 2b) was mainly reflected on PC1 and similarly reflected contrasts between aliphatic hydrocarbons (e.g., 1-Nonene, -Nonadecanone, Octadecane, Nonadecane) and more compounds of unknown origin (referred to as "Peaks 934, 959, 1563, and 1720").

However, we note that the interpretation of individual pyrolysis products is often ambiguous, as detected compounds may originate from different or the same precursor molecules. For this reason, we consider individual peak loadings as difficult to interpret and of limited value to most readers. Instead, we deliberately focused our main analysis on compound-group summaries, which provide a more robust and interpretable representation of SOM composition. Accordingly, all detected pyrolysis products were grouped into SOM compound classes (Supplementary Table S3), and their relative and absolute abundances were analyzed using linear mixed-effects models (Supplementary Figures S3 and S4). Importantly, we want to note that both PCA and compound group abundance patterns describe the same underlying differences in SOM chemistry, but at different levels of resolution.

Figure 4. Were the final community members related to known saprotrophs, endophytes, or ectomycorrhizal fungi? This distribution affects the interpretation of their potential physiological roles. The FungalTraits database may be useful here.

We agree that assigning fungal taxa to functional guilds (e.g., saprotrophic, ectomycorrhizal, endophytic) can provide valuable ecological context for interpreting the observed community patterns. Although functional inference from Arctic ITS1 datasets can be challenging (high taxonomic uncertainty at lower ranks and limited trait coverage is common) we explored the trait-based annotation using the FungalTraits database and will incorporate these data into our revised manuscript.

To balance ecological interpretability and assignment confidence we suggest assignments at the genus level appropriate, as higher taxonomic ranks often encompass multiple or ambiguous ecological strategies. Using this approach, approximately 18% of fungal ASVs could be reliably assigned to functional guilds (whereas 82% remained unclassified). Among the assigned ASVs, the most prevalent guilds were ectomycorrhizal fungi and saprotrophs (including litter-, wood-, and soil-associated taxa), followed by root endophytes. Additional lifestyles (e.g., parasitic, lichenized, or specialized saprotrophic fungi) were present but contributed less than 1% of total ASVs.

This trait-based annotation revealed differences in the distribution of guilds (e.g., ectomycorrhizal fungi) across polygon types and soil layers, which we will incorporate into the revised manuscript to strengthen our ecological interpretations. Nevertheless, we also need to acknowledge that the large majority of taxa could not be reliably assigned, and that overall, rather large uncertainty that is associated with functional inference in Arctic fungal communities. Summary Tables of the FungalTraits database-assignments will be provided via Zenodo Repository.

Section 4.1 The Discussion should be based on interpreting the present Results. Consider omitting text that speculates on tundra processes without connecting to Results. Alternatively, sections like lines 555-560 that discuss potential Archaea-mediated biochemical processes could be bolstered by reference to specific community composition results. See also lines 613-630

We agree that the respective sections could build more on explicit community composition results. We plan to revise the sections and to implement the following changes:

L551-554**:** We will incorporate insights from the FungalTraits analysis, demonstrating that LCP soils (2.2 %) indeed hosted a lower proportion of ectomycorrhizal fungi compared to FCP (7.8 %) and HCP (6.8 %) soils.

Lines 555–560**:** We will highlight archaeal abundance patterns across soil types, and their eminent role in LCP communities. LCP soils accounted for 67% of all ddPCR-corrected archaeal reads in the dataset, compared to 29% in FCP and 4% in HCP soils. Moreover, also the relative abundance of archaea (expressed as a fraction of the total prokaryotic community) was substantially higher in LCP soils (7%) than in FCP (1%) and HCP soils (0.2%). The paragraph will then better connect to the discussion of archaeal-mediated processes in peaty environments.

L555-560: note numerically that archaeal abundance was elevated in LCPs soils compared to FCP and HCP soils. Specifically, LCP soils harbored 67 % of all dd-PCR corrected reads in the dataset that were assigned to archaea (FCP soils 29 %, HCP soils 4 %). Also, the relative abundance of

archaea (expressed as fraction of their total prokaryotic community, respectively) was notably high in LCP (7 %) soils (FCP 1 % and HCP soils 0. 2%).

Lines 613–630: revise the respective paragraph to more tightly connect the discussion of permafrost microbial communities to the results presented in this study. We reduce the general discussion of microbial adaptations to the physical and ecological constraints imposed by the frozen conditions but will connect abundance patterns of specific phyla with those environmental conditions, instead (Supplementary Table 8). For example, we will highlight the strong association of Cloacimonadota, Caldisericota, and Campylobacterota with the permafrost layer, phyla linked to anaerobic organic carbon turnover, including fermentation and sulfur and nitrogen redox processes. In addition, we will refer to the elevated abundance of Firmicutes, which may benefit from dormancy and spore-forming strategies in the freeze–thaw transition zone.

The Conclusions section is unusually long, and it includes significant Discussion text. This section should not introduce any new material; therefore, authors should carefully consider omitting any sentences that include citations. As Josh Schimel recommends in his excellent book "Writing Science" (Oxford) this section should concisely reiterate the from Methods & Results, answer key questions posed in the Introduction, and demonstrate how those answers have advanced understanding of the topic. I suggest shortening and rewriting this section to address those three items.

We thank the reviewer for pointing this out. We will shorten and rewrite the Conclusion section and remove the discussion-style material.

First, we will remove interpretations about potential future trajectories or Arctic lowland tundra ecosystems under climate change, including the topics of polygon transitions, thermokarst, active layer deepening, vegetation change, and rhizosphere priming, and relocate this content to the Discussion section.

Second, we will limit the revised Conclusions to:

(i) briefly restate the importance and strength of the used study approach which considers both major axes of spatial heterogeneity in lowland tundra soils (i.e., polygon types and soil layer) simultaneously.

(ii) directly answering the questions posed in the Introduction, by summarizing that polygon-specific signals persisted across soil layers, and layer-specific patterns were consistent across polygon types, whilst interactive effects seemed comparatively minor. However, we also conclude from the patterns that we observed that most edaphic variability across both spatial scales emerged from the gradients in redox conditions and vegetation-associated organic matter inputs.

(iii) outlining that a very limited number of spatial "units", (i.e., polygon types and soil layers), is sufficient to effectively capture a disproportionate share of edaphic, microbial, and biogeochemical variability in Arctic lowland tundra soils. Therefore, we advocate that differentiating polygon types and major soil layers, with LCPs, and organic topsoils as potential priority units, can provide a

simplified but practical framework for parameterizing and scaling soil processes across this geomorphologically complex ecosystem.

**Minor comments:**

Introduction: It is not necessary to cite the same paper multiple times in a paragraph, particularly in successive sentences.

We acknowledge the comment, and will remove double citations within the same paragraph, accordingly.

Line 62. While HCP and FCP soil may be 'dry' compared to 'LCP' soil, both soils have high SWC compared to typical agricultural or temperate ecosystem soils. This affects diffusion, substrate availability, and microbial growth and should be noted.

We agree that Arctic lowland tundra soils (across all polygon types) generally may all have relatively high soil water content compared to temperate or agricultural soils. Our focus, however, is on relative hydrological differences among polygon types (emerging from their differential microtopography), which strongly regulate oxygen availability, soil organic matter characteristics, microbial community composition, and biogeochemical processes. Certainly, water availability is not a limiting factor for substrate diffusion and substrate accessibility in these soils, but that it's rather the permafrost table that imposes the strongest physical constraint for (L94-95).

When discussing the establishment of predominantly aerobic versus anaerobic microbial communities in drained versus inundated terrain (L78), we also highlight the comparatively lower energy yield of anaerobic respiration compared to aerobic respiration (L545–548), which of course also affects microbial growth rates. We will hence also incorporate the link between microbial growth and redox conditions, accordingly.

Section 2.3. What was learned from the isotopic signature analysis performed here?

Stable carbon and nitrogen isotope composition ($\delta^{13}C$ and $\delta^{15}N$) can provide useful contextual information on soil organic matter (SOM) processing stage and the nitrogen cycle. In our dataset, however, isotopic differences between soil layers and polygon types were relatively subtle and did not reveal strong or consistent trends, which is why we didn't discuss them in the main text of the manuscript. To address the reviewer's comment, we here briefly summarize what can and cannot be inferred from the isotopic data in this study.

Soil $\delta^{13}C$ values showed only minor variation across polygon types and soil layers (from approximately 27 ‰ to –26 ‰). Although vegetation composition differs markedly among LCPs and FCPs/HCPs, but bulk SOM $\delta^{13}C$ values were largely similar likely due to the fact that all plants in the Arctic are C3-plants and differences in water use efficiency are not pronounced. Soil $\delta^{13}C$ values were also not statistically different across soil layers, suggesting that differences in decomposition state were not strongly expressed in bulk SOM $\delta^{13}C$, likely due to extensive cryoturbation activity in this area. This interpretation is also consistent with the largely uniform C:N ratios across layers and confirms that Pyr-GC/MS analyses may reveal finer-scale differences in SOM composition.

Soil $\delta^{15}N$ values may provide contextual information on nitrogen cycling, with higher $\delta^{15}N$ values representing a more open N cycling (N losses) versus lower $\delta^{15}N$ values indicating tighter N cycling (stronger N retention). However, soil $\delta^{15}N$ is a comparatively poor indicator of vegetation-derived imprints and organic matter origin, as site-specific soil properties and multiple interacting edaphic processes produce signals that are difficult to disentangle.

In our study, $\delta^{15}N$ values across polygon types and soil layers were consistently close to 0‰, indicating overall low N losses and a predominantly closed N cycle. This is consistent with the notion of N limitation typical of Arctic tundra ecosystems. In polygon tundra ecosystems, leaching losses are largely restricted by the permafrost table, prolonged frozen conditions throughout the year, and poor drainage due to the flat terrain. Also, gaseous N losses via denitrification are comparatively low, due to limited nitrate availability and the predominance of organic nitrogen in tundra ecosystems. Across polygon types, LCP soils exhibited $\delta^{15}N$ values close to 0‰ (~0.34 ‰), whereas FCP and HCP soils were slightly more enriched (~1.3–1.6 ‰). Organic-rich tundra soils, particularly peaty and waterlogged systems such as LCPs, commonly display $\delta^{15}N$ values near 0‰ while the less organic-rich soils of FCPs and HCPs showed slightly higher $\delta^{15}N$ values. Across soil layers, organic and mineral topsoils showed slightly higher $\delta^{15}N$ values than permafrost, while cryoturbated material exhibited intermediate values. Given the narrow overall range of $\delta^{15}N$ values observed in this study, we believe that these differences should be interpreted cautiously and do not support inferences regarding specific nitrogen sources or transformation pathways. Instead, the $\delta^{15}N$ data primarily constrain the system-level interpretation by indicating generally tight nitrogen cycling with only minor N losses. We therefore consider it important to report $\delta^{15}N$ values in the Supplementary table 1, while deliberately refraining from overinterpreting them.

To better integrate the isotopic results into the revised manuscript, we will (i) explicitly report soil $\delta^{13}C$ and $\delta^{15}N$ patterns in the Results section, and (ii) incorporate a short interpretation into the Discussion. Specifically, we will highlight that while bulk soil C:N ratios and $\delta^{13}C$ values varied little across soil layers, Py-GC/MS analyses revealed more fine-scale differences in SOM quality.

Lines 87-88. What were the bulk densities of the sampled soils? I could not readily find this information by searching the main text or supplemental materials. Did the SIPRE corer cause compression of the soil layers in the thawed LCPs?

Bulk density values have now been added to Supplementary Table 1 and are briefly described in the Results (Section 3.1). These values are also included in the updated CSV dataset deposited on Zenodo. However, please note, that bulk density measurements are missing for five cryoturbated samples, due to the irregular geometry and discontinuous nature of cryoturbated pockets.

Regarding potential compression during coring: yes, the SIPRE corer caused some compression of the active layer in LCPs. To account for the compression during coring, we measured active layer thickness independently using a soil probe, which allowed us to adjust our interpretation of the extracted cores. When field conditions permitted, we also extracted active layer blocks using a spade (Supplementary Fig. 1e) and compared them to our interpretation of cores.

Importantly, our analyses are based on (sample-wise disaggregated) soil horizons rather than absolute depth, so potential active layer compression does not affect our sample classification or conclusions.

Section 2.5 Was the depth of sequence coverage from multiplexed SSU rRNA and ITS2 gene amplicon sequencing sufficient to characterize the community diversity in all samples?

Thank you for the question. As mentioned in the Supplementary Material in L145, we removed samples that were characterized by less than 500 (raw) reads. This specifically meant the removal of 2 samples from the bacterial and archaeal dataset (16S rRNA gene), because of insufficient sequence coverage (2 x FCP_cryoturbated). As well as the removal of 3 samples from the fungal dataset (ITS1 gene region), because of insufficient sequence coverage (1 x FCP_cryoturbated, 1 x FCP_permafrost, and 1 x LCP_permafrost).

We will transfer the section that explains how the amplicon datasets were treated prior to downstream analyses from the Supplement to the Materials and Methods section of the revised manuscript.

Line 510-511. Some discussion of the role of pyrolysis GC-MS results in inferring organic matter quality is warranted here.

We acknowledge the suggestion and will extend these lines accordingly. Specifically, we will state that Pyrolysis-GC/MS is a spectrochemical method that has been successfully applied in permafrost studies (e.g.,Folhas et al., 2025; Verret et al., 2025), providing compound-level details that bulk soil indices may not capture. Indeed, in our study, polygon types showed similar soil C:N ratios and bulk soil $\delta^{13}$C signatures (Supplementary Table 1), while pyrolysis-GC/MS fingerprinting was able to reveal pronounced, but finer-scaled differences in SOM quality among polygon types and soil layers (Fig. 2(a,b)). This highlights the added value of the Pyrolysis-GC/MS method for resolving compositional variability that is not apparent from bulk metrics alone.

References: Citations to non-journal articles like the "Field Book for Describing and Sampling Soils" need to be updated with sufficient information for the reader to find the resource.

Thank you for noticing the mistake. The citation has been updated.

**References:**

Folhas, D., Couture, R.-M., Laurion, I., Vieira, G., and Canário, J.: Natural organic matter dynamics in permafrost peatlands: Critical overview of recent findings and characterization tools, TrAC Trends in Analytical Chemistry, 184, 118153, https://doi.org/10.1016/j.trac.2025.118153, 2025.

Verret, M., Naeher, S., Lacelle, D., Ginnane, C., Dickinson, W., Norton, K., Turnbull, J., and Levy, R.: Preservation and degradation of ancient organic matter in mid-Miocene Antarctic permafrost, Biogeosciences, 22, 5771–5786, https://doi.org/10.5194/bg-22-5771-2025, 2025.